# Flood detection using Gravity Recovery and Climate Experiment (GRACE) terrestrial water storage and extreme precipitation data

**Jianxin Zhang**[1,2]**, Kai Liu**[1]**, and Ming Wang**[1]

[1]School of National Safety and Emergency Management, Beijing Normal University, 100875 Beijing, China
[2]School of Systems Science, Beijing Normal University, 100875 Beijing, China

**Correspondence:** Kai Liu (liukai@bnu.edu.cn)

**Abstract.** A complete global flood event record would aid researchers to analyze the distribution of global floods and, thus, better formulate and manage disaster prevention and reduction policies. This study used Gravity Recovery and Climate Experiment (GRACE) terrestrial water storage and precipitation data combined with high-frequency filtering, anomaly detection and flood potential index methods to successfully extract historical flood days globally between 1 April 2002 and 31 August 2016; these results were then further compared and validated with Dartmouth Flood Observatory (DFO) data, Global Runoff Data Centre (GRDC) discharge data, news reports and social media data. The results showed that GRACE-based flood days could cover 81 % of the flood events in the DFO database, 87 % of flood events extracted by MODIS and supplement many additional flood events not recorded by the DFO. Moreover, the probability of detection greater than or equal to 0.5 reached 62 % among 261 river basins compared to flood events derived from the GRDC discharge data. These detection capabilities and detection results are both good. Finally, we provided flood day products with a 1° spatial resolution covering the range between 60° S and 60° N from 1 April 2002 to 31 August 2016; these products can be obtained from https://doi.org/10.5281/zenodo.6831384 (Zhang et al., 2022b) CE1 . Thus, this research contributes a data foundation for the mechanistic analysis and attribution of global flood events.

## 1 Introduction

Flood disasters threaten the lives of millions of people around the world every year, causing more economic loss than any other natural disaster. An increasing number of extreme weather events are occurring more frequently under global climate change (Schinko et al., 2017). The latest research on global flood disasters shows that the proportion of the population at risk from floods is increasing each year (Tellman et al., 2021).

Existing global flood data mainly include historical data records, hydrological model simulations and remote sensing observations. Historical data records include those of the Dartmouth Flood Observatory (DFO) (Brakenridge, 2022), the international disasters database (EM-Dat) (Guha-Sapir et al., 2021), Munich Re's NatCatSERVICE (https: //natcatservice.munichre.com, last access: 6 June 2022), and Sigma (Swiss Re, 2022). The DFO database mainly records large-scale flood events from news reports and from governmental, instrumental and remote sensing sources. This database records not only the country, latitude, approximate scope, and start and end time of each event but also the cause and severity level of the event. There have been approximately 4700 major flood events since 1985. EM-Dat contains basic core data on the occurrence and impact of more than 22 000 large-scale disasters in the world from 1900 to the present. This database is compiled from a variety of sources, including United Nations (UN) agencies, nongovernmental organizations, insurance companies, research institutes and news organizations. NatCatSERVICE is a natural hazard-based disaster loss database with up to 28 000 entries owned by Munich Re. Sigma is also a global disaster database com-

prising anthropogenic and natural catastrophe losses since 1970. In addition to recording basic disaster information, this database includes the total and insured losses. Flood data are also derived from global hydrological models. For example, the University of Maryland's Global Flood Monitoring System (GFMS) takes real-time integrated precipitation information (Tropical Rainfall Measuring Mission, TRMM, and Global Precipitation Measurement, GPM, data) as inputs in a quasi-global (50° N–50° S) hydrological runoff and routing model to run 1/8° gridded data. Surface water storage statistics are used to derive flood thresholds for each grid location, and the depth above the corresponding threshold is calculated as the flood intensity (Wu et al., 2014, 2012a, b, 2011). Another example is the Floods.Global system (http://floods.global, last access: 6 June 2022) database, in which Integrated Multi-satellite Retrievals for GPM (IMERG) precipitation data are used to estimate future 72 h flows, with coverage from 60° N to 60° S and a resolution of 0.1°. With the development of remote sensing satellite products in the 1980s, a cost-effective flood monitoring method emerged. As long as there are historical areas over which satellites have passed and imaged, there are opportunities to observe flood events; these methods are more realistic and effective than flood models with respect to characterizing actual observed flood areas. Commonly used remote sensing data include optical remote sensing images and microwave remote sensing images, among which microwave remote sensing technologies, especially the commonly applied synthetic aperture radar (SAR), are used. Rättich et al. (2020) developed an automatic procedure to evaluate flood durations and uncertainties using multiple satellites, including Sentinel-1, Sentinel-2, Landsat 8 and TerraSAR-X. The method was successfully demonstrated on the 2019 flood in Sofala Province, Mozambique, and on the 2017 flood in Bihar, India. Tellman et al. (2021) used 250 m resolution Moderate Resolution Imaging Spectroradiometer (MODIS) data to extract the inundation extents of a total of 913 global flood events from 2000 to 2018, thereby providing data support for vulnerability assessments and flood model improvements. Tong et al. (2018) used both Landsat 8 optical imagery and COSMO-SkyMed radar imagery combined with a support vector machine and the active contour without edges model to perform flood monitoring. The results showed high accuracies of 97.46 % for optical imagery and 93.70 % for radar imagery.

However, there are some limitations in current databases. NatCatSERVICE, EM-Dat and Sigma provide data only at the country level. The NatCatSERVICE database covers most large flood events around the world but only a few small flood events in developing countries due to restricted connectivity (de Bruijn et al., 2019). Although there are approximate map locations available in the Sigma and NatCatSERVICE databases, specific location names are not publicly available (Moriyama et al., 2018). Moreover, both the NatCatSERVICE and Sigma databases are developed by reinsurance companies, and the accessibility of the information

in these databases is limited (Moriyama et al., 2018; Kron et al., 2012; Huggel et al., 2015). The EM-Dat database records only the numbers of flood events in different countries without corresponding spatial location information. Although DFO records the start and end times as well as the approximate spatial locations of flood events, the duration is sometimes long (more than 1 or 2 months), and the spatial locations are only roughly delineated according to news reports. Tellman et al. (2021) extracted flood extents and analyzed the population exposure of 913 large-scale flood events from 2000 to 2018 based on MODIS daily data with a resolution of 250 m, thereby finely delineating the spatial inundation extent. During this period, there were more than 3000 flood events recorded in the DFO database, whereas the number of MODIS-derived floods was less than 30 % of that recorded by DFO. The numbers of flood events recorded in China, Russia and Canada are obviously lacking. Moreover, flood detection methods based on remote sensing data are mainly aimed at specific flood events in small areas and are influenced by the number of revisit cycles at the same location (especially for SAR images) and bad weather (especially for optical images) (Kussul et al., 2011; Hostache et al., 2018; Manavalan, 2017). The spectral information of optical remote sensing images is influenced by clouds, affecting the quantitative inversion of flood extent based on remote sensing. SAR images lack revisits of the same location for flood change detection. These shortcomings affected the flood extraction accuracy. There is a need to fill in the missing flood events with a new observational dataset.

Another remote sensing technique based on gravity satellites, the Gravity Recovery and Climate Experiment (GRACE), has also been successfully used to detect flood events. Reager and Famiglietti (2009) first proposed the use of the terrestrial water storage capacity and flood potential index, creating a precedent for GRACE to assess large-scale flood events. This method was subsequently improved upon and applied to different river basins. Molodtsova et al. (2016) found an agreement between the flood potential index derived from GRACE and recorded floods by using multiyear flood observation data from 2003 to 2012 from the United States (US) Geological Survey and DFO. Gupta and Dhanya (2020) proved that GRACE terrestrial water storage (TWS) and the flood potential index had the capability to assess hydrological extreme events over heterogeneous regions with the occurrences of high-intensity and long-duration floods. They suggested that flood potential index can be useful for flood monitoring when discharge data are rarely available. With the continuous progress of the global GRACE-only gravitational field solution, GRACE daily data products have also been effectively developed and applied (Kvas et al., 2019; Mayer-Gürr et al., 2018). Gouweleeuw et al. (2018) used a daily solution based on GRACE TWS and daily river runoff data to assess major flood events in the Ganges–Brahmaputra Delta and confirmed the method's potential for gravity-based large-scale flood monitoring. Xiong

et al. (2022) used daily downscaled GRACE data to detect short-duration and high-intensity floods. They found that there was a strong correlation between the high-frequency components of GRACE TWS and runoff.

In this study, we focus on extracting global historical flood events based on daily GRACE and precipitation data. Apart from being affected by battery management in some months, GRACE's gravity measurements cover most months and are not affected by varying weather conditions. This study mainly extracted all flood days in the historical time series caused by extreme precipitation, regardless of whether the flood event caused severe damage. Finally, we provided global flood days with a resolution of 1° during the period from 1 April 2002 to 31 August 2016. These data replenish the missing flood events in the historical record and provide a new and complete flood dataset, thereby contributing a sufficient data foundation for research on the inducement of global floods.

## 2 Data

### 2.1 Daily GRACE TWS

The GRACE constellation is a pair of twin satellites that can measure changes in Earth's gravitational field. There is a precise radar rangefinder between the two satellites. When Earth's gravitational field changes slightly, it can be detected by either of the two satellites. The distance signal between the two satellites is amplified to measure the state change at the current moment relative to the previous moment (Cazenave and Chen, 2010). The short-term gravitational field changes of Earth are mainly caused by changes in factors such as terrestrial water storage, atmospheric water vapor and ocean tides. When these signals are deducted, the change in the entire terrestrial water storage can be inverted (Wahr et al., 1998). The daily GRACE data selected in this study come from daily solutions obtained using Kalman smoothing by Mayer-Gürr et al. (2018), Graz University of Technology, based on the ITSG-Grace2018 gravity field model. The ITSG-Grace2018 gravity field model, which offers unconstrained monthly and Kalman-smoothed daily solutions, is the most recent GRACE-only gravity field model computed in Graz (Mayer-Gürr et al., 2018). The time period spans from 1 April 2002 to 31 August 2016, the resolution is 1° and the unit is meters (m). A third-order autoregressive (AR) model was used to stabilize the daily solution. A set of spherical harmonic coefficients for the various degrees ($n = 2 \ldots 40$) was estimated. When GRACE data were not available for a specific day, daily solutions were delivered through an adjustment process (Bergmann-Wolf et al., 2015; Dill, 2008). These processed data can be obtained from the following website: https://www.tugraz.at/institute/ifg/downloads/gravity-field-models/itsg-grace2018/ (last access: 21 November 2021).

### 2.2 Precipitation

This study used Global Precipitation Measurement (GPM) data to calculate extreme precipitation. GPM is an international satellite mission launched by the National Aeronautics and Space Administration (NASA) and Japan Aerospace Exploration Agency (JAXA). It is the next-generation, high-quality global rain and snow satellite observation network after the TRMM. GPM provides an important data foundation for scientific researchers to understand the Earth's water resources and energy cycles and improve their ability to predict extreme events (Huffman et al., 2019). The resolution of these data is 0.1°, with the unit of millimeters (mm), mainly covering the range of 60° S–60° N, and both north–south latitudes of 60–90° have partial coverage. This study selects the IMERG Final Run product, which uses global microwave precipitation data, infrared data, precipitation station data and other potential precipitation indicators to cross-calibrate, fuse and interpolate TRMM and GPM data at refined temporal and spatial scales. It is an officially recommended product and can be obtained from the following website: https://gpm.nasa.gov/data/directory (last access: 10 October 2021). To remain consistent with the GRACE resolution and maintain extreme precipitation signals, we take the maximum values of the precipitation covered by each 1° GRACE grid to further calculate the flood potential index and the number of extreme precipitation days.

### 2.3 Flood events from Dartmouth Flood Observatory

The DFO dataset records large flood events from various news reports as well as governmental, instrumental and remote sensing sources. It contains the start and end times of each flood, the country where it occurred, the approximate flood extent, the cause of the flood and the degree of damage. It is a rare and useful product for studying global historical floods. This data product has been widely used in flood hazard science research (Tellman et al., 2021; Hagen et al., 2010; Winsemius et al., 2013; Idowu and Zhou, 2019). This study focuses on precipitation-induced floods. A total of 2380 flood events in the 60° S–60° N range were caused by heavy precipitation between 1 April 2002 and 31 August 2016. This product was primarily used to validate the flood data extracted in this study and can be obtained from https://floodobservatory.colorado.edu/Archives/index.html (last access: 1 October 2021).

### 2.4 MODIS-derived flood inundation data

The flood inundation extent data used in this study come from a total of 807 flood events extracted based on MODIS data by Tellman et al. (2021) in the 60° S–60° N region from 1 April 2002 to 31 August 2016. This product was produced based on atmospherically corrected Terra (MOD09GA/GQ) and Aqua (MYD09GA/GQ) MODIS images. The authors then used threshold analysis methods (including standard

and Otsu-optimized threshold methods) and slope constraints (slopes greater than 5° were masked out) to extract inundations at a 250 m spatial resolution according to the flood events recorded by the DFO (Tellman et al., 2021). The MODIS-based floods were compared and verified for coincidence with the 30 m resolution inundation data derived from Landsat 5, 7 and 8 images, and flood map quality control analysis was also performed. This product relies on the Google Earth Engine platform (Gorelick et al., 2017), which can be obtained from the following site: https://developers.google.com/earth-engine/datasets/catalog/GLOBAL_FLOOD_DB_MODIS_EVENTS_V1 (last access: 1 October 2021). These data were further refined on the basis of DFO-recorded approximate flood extent and provided a reliable spatial inundation dataset for verification in our study.

## 2.5   Global Runoff Data Centre (GRDC) discharge data

The Global Runoff Data Centre is an international data center operating under the auspices of the World Meteorological Organization. It was established in 1988 to support research on global climate change and integrated water resource management. We downloaded the global mean daily discharge data from https://www.bafg.de/GRDC/EN/Home/homepage_node.html (last access: 2 November 2021), which additionally contained other attributes, like the country, longitude, latitude and river name, associated with each flood event. The unit of mean daily discharge is cubic meters per second ($m^3 s^{-1}$), and the stations with more than 50 % of days missing in the research time period (1 April 2002–31 August 2016) were excluded to ensure accuracy. Finally, we obtained 3408 stations from 1 April 2002 to 31 August 2016 as the validation dataset to verify the GRACE-derived flood days.

## 3   Methods

Figure 1 shows the technical workflow of this study. It mainly consists of data preparation, extraction flood days and result verification. Daily precipitation and daily GRACE TWS data are used for the flood data extraction step, and daily discharge, DFO, MODIS-derived flood inundation and social media data are used for the flood validation step. The flood extraction step is mainly based on high-frequency signals of TWS and the flood potential index to obtain the preselected possible flood days; extreme precipitation constraints are then used to obtain the final flood days. The flood validation includes comparisons with the DFO-recorded flood extent, MODIS-derived flood inundation, GRDC discharge-derived flood events and significant flood events recorded on social media.

## 3.1   Seasonal and trend decomposition using loess (STL)

Extreme precipitation has sudden characteristics, and its signals are reflected in the high-frequency signals of GRACE (Xiong et al., 2022; Gouweleeuw et al., 2018). Seasonal and trend decomposition using loess (STL) (Robert et al., 1990) is a filtering process as well as a general and robust time series decomposition and forecasting method used to decompose time series variables into seasonal, trend and remainder components for further forecasting. This process can handle data with any type of seasonality as well as high-frequency signal data. It also allows seasonal components to vary over time and is robust to outliers. In this study, we selected this method as a high-pass filtering tool to process GRACE TWS and obtain high-frequency signals (excluding seasonal and trend components) for subsequent analyses. In this work, the STL function in the R language "stats" package was used to process all grid time series corresponding to the GRACE TWS period (1 April 2002–31 August 2016). The two main parameters, "t.window" and "s.window", should be specified when using STL. "t.window" is the number of consecutive observations when estimating the trend cycle; it was set to a 31 d window to cover the month and separates daily data according to Gouweleeuw et al. (2018) and Xiong et al. (2022). "s.window" is the number of consecutive years when estimating each value in the seasonal component; it was set to 360, which was determined using a Fourier transform to convert to the frequency domain to obtain the frequency corresponding to the maximum amplitude.

## 3.2   Anomaly detection based on a generalized extreme studentized deviate test

The generalized extreme studentized deviate (GESD) test (Rosner, 1983) is a simple and effective statistical method for detecting one or more outliers in univariate data that follow an approximately normal distribution. It has been widely used in the field of hydrological anomaly detection (Saghafian et al., 2014; Clark and Zipper, 2016). The GESD test is mainly used in this study to extract possible flood days corresponding to the high-frequency signals. In this study, the method selected for extracting flood information from the high-frequency signals needed to ensure minimum impact from the random error in the high-frequency signal and maximum flood signal extraction. The method requires only that an upper bound for suspected outliers be specified and determines the number of possible outliers based on hypothesis testing (Rosner, 1983). The basic assumptions of GESD are as follows:

there are no outliers in the dataset ($H_0$);

there are at most $r$ outliers in the dataset ($H_a$).

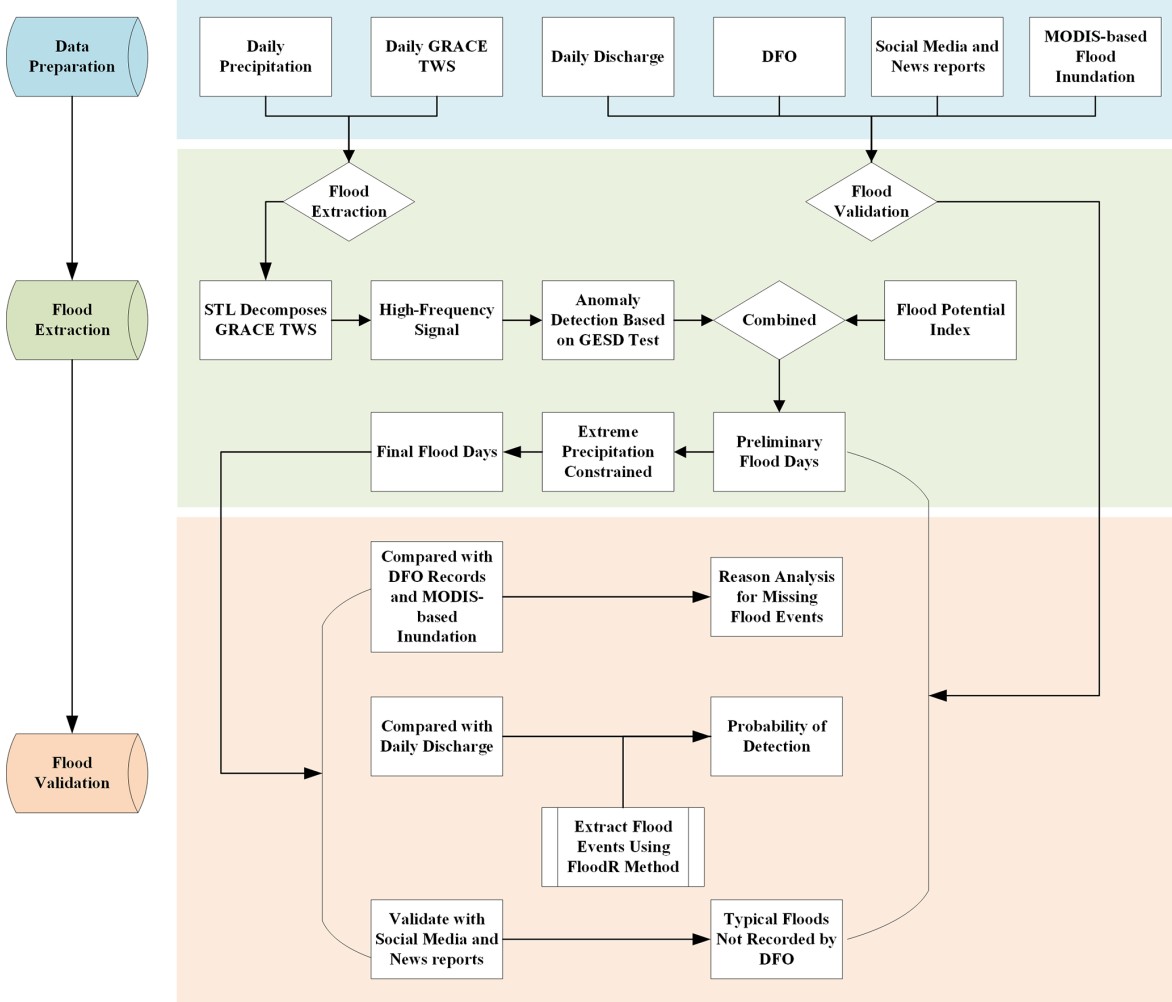

**Figure 1.** Workflow of the global flood extraction approach based on GRACE and precipitation data.

The corresponding statistic is calculated as follows:

$$R_i = \frac{\max_i |x_i - \bar{x}|}{\sigma}, \tag{1}$$

where $\bar{x}$ and $\sigma$ are the sample mean and standard deviation, respectively. After each iteration in which the largest $|x_i - \bar{x}|$ value is removed, the remaining statistics are calculated, and the above process is repeated until at most $r$ outliers are removed.

Consistent with the $r$ test statistics, the $r$ critical values are computed as follows:

$$\lambda_i = \frac{(n-i)t_{p,n-i-1}}{\sqrt{(n-i-1+t_{p,n-i-1}^2)(n-i+1)}}; \quad i = 1, 2, \ldots, r; \tag{2}$$

$$p = 1 - \frac{\alpha}{2(n-i+1)}. \tag{3}$$

Here, $t_{p,v}$ is the $100p$ percentage point in a $t$ distribution with $v$ degrees of freedom, and $\alpha$ is the significance level.

The number of final outliers is then determined by the corresponding maximum $i$ value in $R_i > \lambda_i$ (Rosner, 1983).

Considering that the GRACE high-frequency signal contains both random errors and useful signals, we used the GESD test to control the number of outliers so that they were not affected by subjective thresholds. In this study, the "AnomalyDetection" package (https://github.com/twitter/AnomalyDetection, last access: 2 November 2021) (Aggarwal, 2013; Chandola et al., 2009; Rosner, 1983; Vallis et al., 2014) was used to extract GRACE high-frequency signal data. This package not only includes the GESD algorithm but can also specify the direction of detected outliers. The parameter "direction" indicates whether to extract peaks or valleys, where "pos" means the extraction of peaks, and "neg" means the extraction of valleys. As we considered extreme weather events caused by heavy precipitation in this study, important information was contained in the peak. The main parameter "direction" was set to the "pos", and the maximum possible number of abnormal days "max_anoms" was set to

0.1 to cover the maximum number of abnormal days among the global time series comprising every grid. We provide an example in Fig. S1 in the Supplement to show the process of extracting possible flood days from high-frequency signals of GRACE TWS using the GESD test method as well as the reason for missing some flood events. We have also considered the reliability of the GESD test method in Fig. S2.

### 3.3 Flood potential index

We used the probable flood days extracted by the flood potential index (FPI) to supplement the inability of GRACE to detect flood events with high-frequency signals. The FPI mainly considers rainfall-induced floods and has been widely used to evaluate flood events (Gupta and Dhanya, 2020; Molodtsova et al., 2016; Reager et al., 2014). Its basic assumption is that the regional water storage capacity can be approximated by the maximum value of historical TWS time series. The water storage capacity at the current time can be calculated by subtracting the TWS at the previous time from the maximum value of TWS time series. The proposal of this method was based on monthly data, but this does not affect its application to GRACE daily data. The application of the FPI was undertaken as follows.

The water storage capacity of the current day can be expressed as the temporal difference between the maximum time series value and the previous day's value; the formula is expressed as follows:

$$\mathrm{TWS_{DEF}}(t) = \mathrm{TWS_{MAX}} - \mathrm{TWS}(t-1), \qquad (4)$$

where $\mathrm{TWS_{DEF}}(t)$ represents the maximum allowable relative water storage change on the current day, $\mathrm{TWS_{MAX}}$ represents the maximum value over the entire time series and $\mathrm{TWS}(t-1)$ represents the TWS value of the current day relative to the previous day. A low storage deficit and high precipitation result in a high probability of flooding, i.e., the occurrence of floods should be based on the mismatch between the extreme precipitation level and the increase in water storage, as follows:

$$F(t) = P_{\mathrm{day}}(t) - \mathrm{TWS_{DEF}}(t), \qquad (5)$$

where $P_{\mathrm{day}}(t)$ represents the daily precipitation, and $F(t)$ represents whether the current precipitation matches the water storage capacity. When $F(t) > 0$, flooding may occur. This study uses the FPI to supplement possible flood days in the case that the daily GRACE TWS data have lost useful high-frequency signals due to the interpolation process. In Fig. S3, we have provided an example in which the FPI was able to supplement some flood events not identified by GRACE high-frequency signals.

### 3.4 Flood detection based on GRACE TWS and precipitation data

The flood extraction mainly went through a preselection stage and a final selection stage. We first used GRACE high-frequency signal data combined with the GESD method and FPI to preselect the possible flood days pixel by pixel. Next, we further used the number of extreme precipitation days to constrain and obtain the final flood days. This study focuses on flood events caused by heavy precipitation. Considering that floods are caused not only by single-day precipitation but also by cumulative precipitation, we calculated the extreme precipitation days based on the 1 d precipitation, 3 d cumulative precipitation and 5 d cumulative precipitation. Regarding extreme precipitation, the most commonly used metric is the percentiles of the precipitation time series data, including the $n$th quantiles of the entire time series data; alternatively, the $n$th quantile of wet days (daily precipitation $> 1$ mm) can be considered (Myhre et al., 2019; Pendergrass, 2018; Shi et al., 2021). In order to present the calculation process more clearly, we randomly selected a spatial grid for detailed processing. Figure S4 shows the intermediate process of flood day extraction. This process was carried out as follows:

1. The high-frequency signal of the TWS was extracted using the STL method.

2. The possible flood days were calculated using the GESD method.

3. We used the FPI to supplement possible flood days in the case that the daily GRACE TWS data had lost useful high-frequency signals due to the interpolation process.

4. We constrained the preselected floods using the extreme precipitation days derived from daily and cumulative precipitation. Based on the principle of extracting as many flood events as possible with as few errors as possible, we choose the 95th quantile of the entire time series as the condition to constrain the flood days extracted from GRACE data. We also provide GRACE-based flood days obtained with the 90th and 99th quantiles of the entire time series data and wet days at https://doi.org/10.5281/zenodo.6831384 (Zhang et al., 2022a). TS1

### 3.5 Flood event extraction based on daily discharge data

To verify the reliability of the extracted results, this paper used the global discharge data products released by GRDC and the statistics-based automated flood event extraction (FloodR) method to extract possible flood events. FloodR is a statistics-based flood event separation method proposed by Fischer et al. (2021). It can automatically separate flood events using a univariate daily discharge time series, and it

includes additional tool for manually checking and correcting the separation results quickly, allowing expert knowledge to be easily incorporated. Considering that the fluctuation in daily discharge data is smoother than that in hourly discharge data, FloodR used the moving-window variance to overcome the lower dynamic characteristics of daily discharge. Its basic rules include three points: (1) a flood event is an event that temporarily exceeds the normal discharge, and the start and end of each flood event can be defined; (2) a flood event can be characterized by significantly increased dynamics of discharge; (3) the sum of the increasing discharges is similar to the sum of the recession of the flood event (Fischer et al., 2021). FloodR can also automatically handle missing data and perform flood separation in segments according to the missing data before finally merging them. In this paper, the "eventsep" function in the FloodR package was used and the parameters were set to default (according to the practice of Fischer et al., 2021 CE2), while the "NA_mode" parameter was based on whether there were missing values in the discharge time series. The results extracted by FloodR include information like the start and end times of each flood, the flood peak date, and the flood baseflow, thereby providing an important data foundation for verifying the time series comparison ability of this study.

We use the goodness of flood separation (GFS) to evaluate the performance of the FloodR method. This indicator explicitly minimizes the number of small runoff events and maximizes the number of flood events with high discharge. This indicator can be used to address the lack of a consistent and true data foundation for the evaluation of the goodness of flood separation (Fischer et al., 2021).

$$\text{GFS} = \left( \frac{Q_{Q > \text{TH}_{\text{upper;Flood}}}}{Q_{Q > \text{TH}_{\text{upper}}}} \right) - \max \left( \frac{Q_{Q > \text{TH}_{\text{lower;Flood}}}}{Q_{Q > \text{TH}_{\text{lower}}}} - \text{Tol}_{\text{lower}}, 0 \right), \tag{6}$$

TS2 where $Q_{Q > \text{TH}_{\text{upper;Flood}}}$ is the number of flood days with discharge above the threshold of $\text{TH}_{\text{upper}}$, $Q_{Q > \text{TH}_{\text{upper}}}$ is the number of days above the threshold of $\text{TH}_{\text{upper}}$, $Q_{Q > \text{TH}_{\text{lower;Flood}}}$ TS3 is the number of flood days with discharge below the threshold of $\text{TH}_{\text{lower}}$ and $Q_{Q > \text{TH}_{\text{lower}}}$ TS4 is the number of days below the threshold of $\text{TH}_{\text{lower}}$. The upper threshold $\text{TH}_{\text{upper}}$, lower threshold $\text{TH}_{\text{lower}}$ and tolerance threshold $\text{Tol}_{\text{lower}}$ are set as the 95th quantile of discharge, the 50th quantile of discharge and 1 % of the discharge days below the lower threshold, respectively, according to the suggestion of Fischer et al. (2021). CE3

## 3.6 Probability of detection (POD)

In order to better compare the relationship between flood events (observed from DFO, MODIS and discharge) and flood days (derived from GRACE), we referred to the probability of detection (POD) index proposed by Yang et al. (2021) and made it more appropriate for our study.

$$\text{POD} = \text{flood}_{\text{GRACE-based}} / (\text{flood}_{\text{observed}} + \text{flood}_{\text{miss}}), \tag{7}$$

where $\text{flood}_{\text{GRACE-based}}$ denotes flood events identified by GRACE, and $\text{flood}_{\text{observed}}$ denotes DFO-recorded flood events, MODIS-derived flood events or discharge-derived flood events. If each flood event with a 3 or 5 d buffer could cover the GRACE-based flood days, we consider it a $\text{flood}_{\text{GRACE-based}}$ event.

# 4 Results

## 4.1 Flood days and events based on GRACE TWS and precipitation data

This study considers GRACE-based flood days obtained under the constraint of the 95th percentile of the entire time series dataset. Figure 2 shows the global cumulative flood days and flood events from 1 April 2002 to 31 August 2016, and Fig. 3 shows the histograms of flood days and flood events corresponding to Fig. 2. Although the number of flood days extracted above cannot accurately reflect how many flood events occurred, we can simplify the results such that a number of consecutive detected flood days or the interval between 2 consecutive flood days, no more than 3 d or 5 d can be considered a flood event. The principle involves roughly calculating the spatial distribution of global flood event occurrences. Consistency was found between the global spatial distributions of flood events and flood days. We found that 99.8 % of the grids around the world experienced fewer than 400 flood days except in Southeast Asian countries and countries at the junction of North and South America, which experienced more than 400 flood days from 1 April 2002 to 31 August 2016. In addition, the areas with the most flood days and events were mainly located in the tropics. Island countries, western Africa, India, the Himalayas, southern China, etc. were also prone to floods. From the perspective of the divided flood events, the number of grid cells with fewer than 100 events accounted for 96.15 % (Fig. 2b), 97.11 % (Fig. 2c) and 97.52 % (Fig. 2d) of all grids.

We also calculated the average flood days in the same month in each year from 2002 to 2016 to identify seasonal characteristics. As shown in Fig. 4, the global flood distribution reflected obvious seasonal characteristics, and differences between the Northern and Southern hemispheres are clear. More flood days were identified in the Northern Hemisphere in summer (approximately June–September), while flood days in the Southern Hemisphere were concentrated from December to March.

## 4.2 Flood days in mountain glacier regions

We also analyzed the distribution of flood days in the mountain glacier regions. We used global glacier outline data from the Randolph Glacier Inventory (RGI). This dataset can be

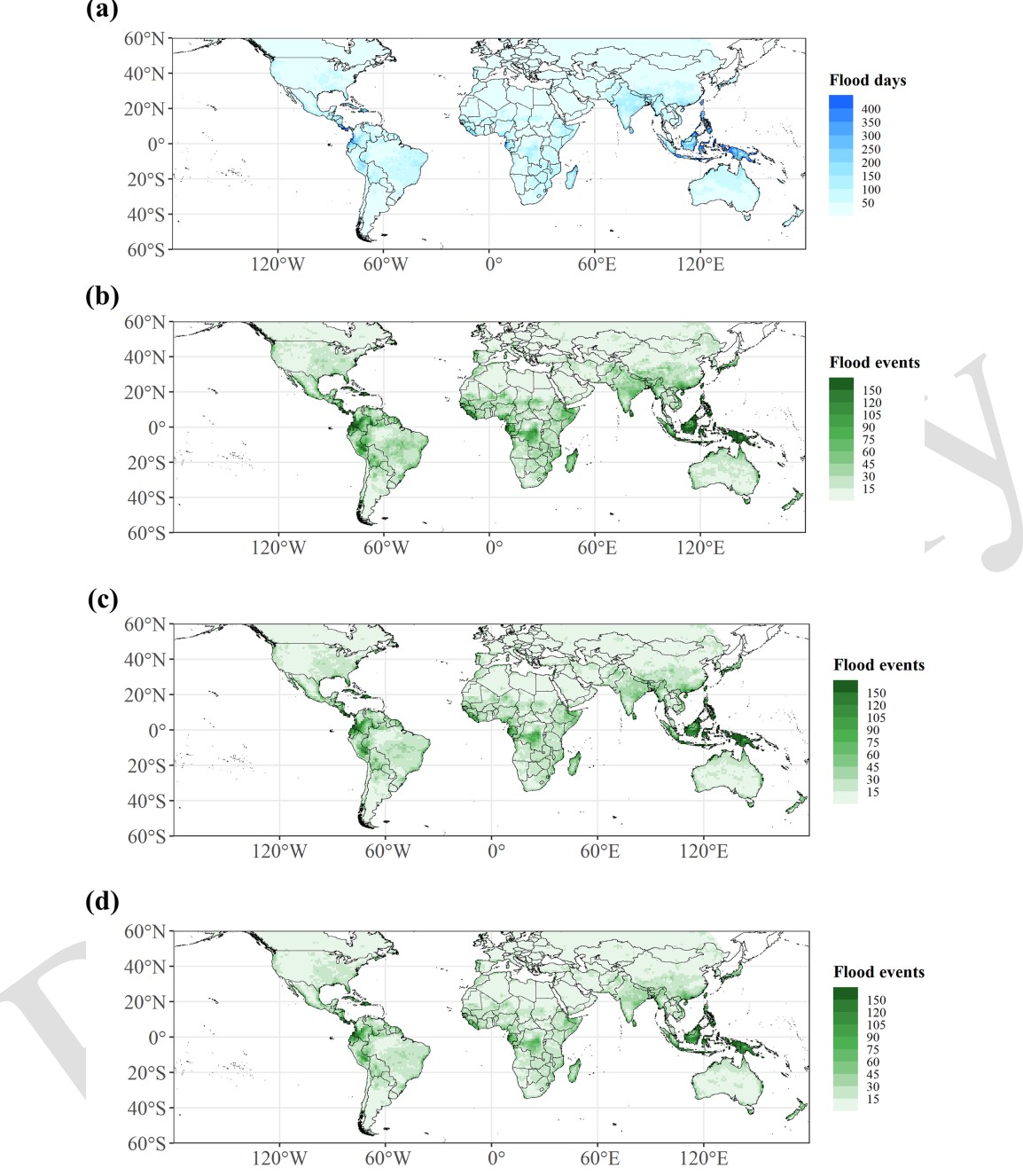

**Figure 2.** Spatial distribution of global floods: **(a)** global flood days from 1 April 2002 to 31 August 2016; **(b)** global flood events based on consecutive flood days; **(c)** global flood events based on the interval between 2 consecutive flood days not exceeding 3 d; **(d)** global flood events based on the interval between 2 consecutive flood days not exceeding 5 d.

used to estimate glacier volumes, rates of elevation change at regional and global scales, and the response of the cryosphere to climate forcing. The dataset is updated annually in shapefile format. In this paper, we used it to locate global glaciers (RGI Consortium, 2017).

In the range of 60° S–60° N, there are 10 glacial regions and 163 flood events recorded based on the DFO database (Fig. 5). A total of 142 flood events were identified, and 21

flood events were not detected, resulting in a POD of 0.87. The capacity of flood detection is close to the global POD (0.81). The results showed that GRACE also has good potential with respect to identifying precipitation-induced floods in glacial regions. Of these 21 flood events, 4 flood events could not be identified due to missing months in GRACE data. Eight flood events had a maximum daily precipitation of less than 40 mm according to the DFO-recorded time period

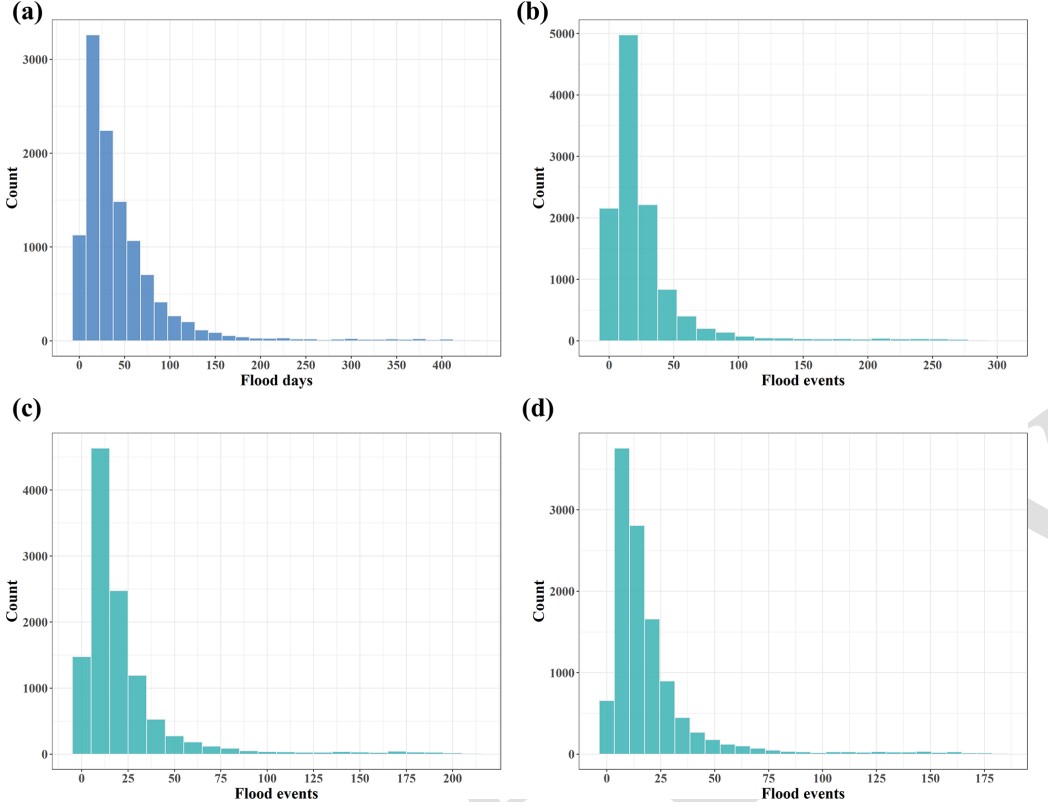

**Figure 3.** Histograms of cumulative flood days and flood events from 1 April 2002 to 31 August 2016: **(a)** histogram of global flood days from 1 April 2002 to 31 August 2016; **(b)** histogram of global flood events based on consecutive detected flood days; **(c)** histogram of global flood events based on the interval between 2 consecutive flood days not exceeding 3 d; **(d)** histogram of global flood events based on the interval between 2 consecutive flood days not exceeding 5 d.

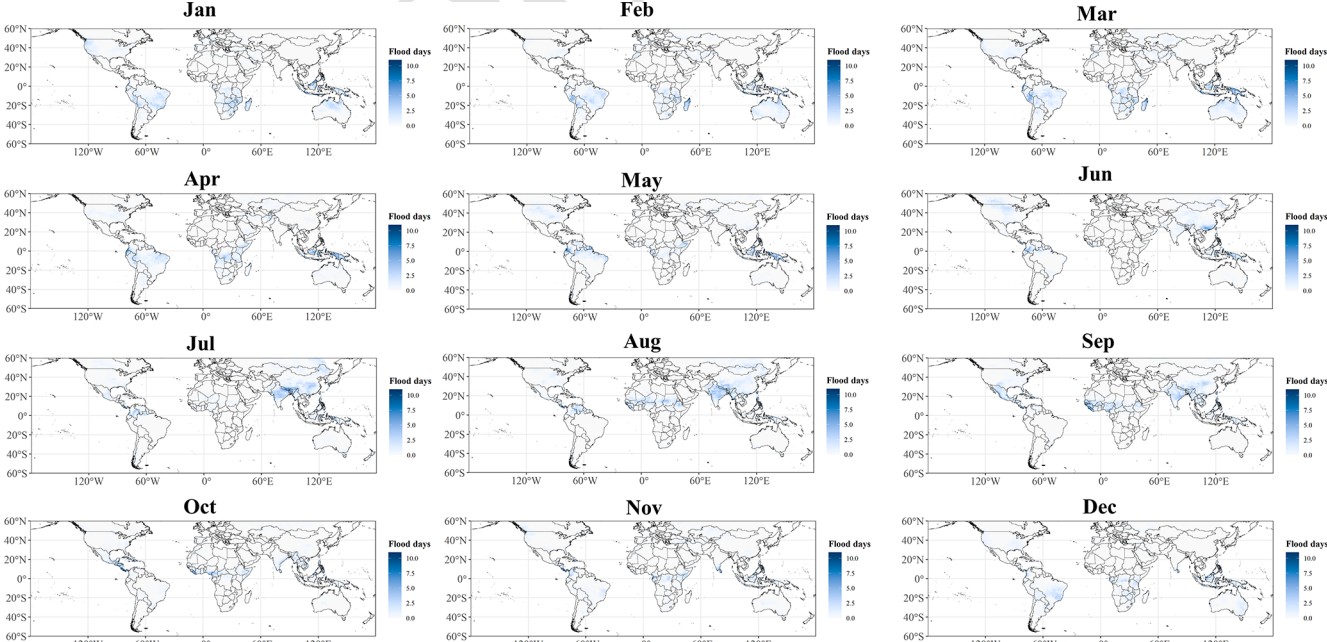

**Figure 4.** Average flood days in the same month of each year from 2002 to 2016.

and spatial location (minimum of 8.44 mm and maximum of 36.56 mm), and GRACE could not identify the weaker signal. The remaining nine flood events could not be identified due to GRACE itself failing to identify flood conditions.

We further selected the GRACE grid covering the glacial regions and analyzed the characteristics of the extracted flood days. Figure 6 shows the 10 detailed glacial regions and the corresponding selected GRACE grids that covered the main glacial areas. Figure 7 shows the results of the extracted flood days related to the grid of each region. In general, the number of flood days in the glacial regions was relatively small, and flood days were mostly concentrated within 50 d from 1 April 2002 to 31 August 2016, whereas the glaciers in the South Island of New Zealand and the glacial regions in the east of southern Asia exceeded 100 d. The areas of the South Island of New Zealand where mountain glaciers are located experience a hyper-maritime climate, and the west coast of the South Island receives the most precipitation (annual precipitation > 12 m) (Anderson et al., 2010). Glacial regions in eastern South Asia are mainly located in the Himalayas, where normal climatic fluctuations become rather quick in the Himalayan sectors due to topography and the southwest Indian Ocean monsoon. Cloud bursts, high winds, snowstorms, etc. can also cause quick floods (Nandargi and Dhar, 2011).

## 4.3  Validation with DFO and MODIS-derived flood data

Figure 8 shows the spatial distribution of 2380 precipitation-type floods recorded by the global DFO from 60° S to 60° N. Judging by the floods recorded by the DFO, floods have occurred in most parts of the world except in the Sahara Desert, the Great Victoria Desert and the northern part of North America.

In this study, the temporal length of the DFO database was compared with GRACE-based flood data throughout the 1 April 2002–31 August 2016 period. According to the database attributes, flood events caused by heavy precipitation were extracted as the validation dataset for this study. Given that the temporal and spatial DFO recording characteristics are approximate and considering the effect of advanced or delayed times on short-duration records, the start and end times of the DFO records were extended forward and backwards by 3 or 5 d, respectively, when being compared with the flood day results. Similarly, when the extent of the DFO polygon was less than 3°, we appropriately built a buffer (3°) to compensate for the positioning errors. We then detected every event in the DFO record to determine whether flood days could be identified based on its temporal and spatial coverage.

Figure 9 shows the distribution of the number of flood events recorded by DFO on 1° spatial grids (the same as the GRACE spatial resolution). It also shows that the eastern part of North America, the northern part of South America, the central and southern parts of Africa, western Europe,

northern India and southern China are all areas with high-frequency flood events. Except for the archipelagic countries in Southeast Asia, the entire spatial distribution is consistent with our results.

We compared the 2380 precipitation-type flood events recorded by the DFO one by one with the real flood extent extracted by Tellman et al. (2021) based on remote sensing images. Figure 10 shows part of the flood event comparison results derived based on GRACE, MODIS and DFO data. The dark blue polygons show the approximate flood ranges delineated by the DFO, red pixels are the flood inundation areas extracted based on MODIS data, and light blue regions show the flood days ($\geq 1$ d) extracted using GRACE TWS and extreme precipitation data during the period recorded by the DFO. MODIS-based inundation extents were calculated according to the DFO time period as well as the union of DFO polygons and HydroSHEDS Basins Level 4 data (Tellman et al., 2021; Lehner et al., 2008; Lehner and Grill, 2013). We also used the DFO-recorded time series as a reference to filter flood days in each grid cell and obtained the spatial flood distributions in specific areas. The flood extents recorded by DFO are rough, and time durations are sometimes long (much more than 1 month); this causes large uncertainties in the spatial distribution and duration. Although the MODIS resolution (1 km) is higher than that of GRACE TWS (1°, $\sim 100$ km), only a limited amount of flooding can be identified by remote sensing images due to the influence of bad weather. The GRACE-based flood days that we provided were only able to indicate the presence of flood events under the $\sim 100$ km grid coverage, and specific detailed flood extents require further identification using high-resolution satellite remote sensing images. Therefore, we note some differences among the spatial patterns of floods recorded by DFO, MODIS and GRACE. This study focuses on whether the flood events recorded by the DFO can be detected by GRACE; as long as the number of flood days ($\geq 1$ d) extracted by GRACE could be found at the time and in the space specified by DFO, the effectiveness of the method could be demonstrated. Considering the results of 2380 events, 463 flood events were not detected, resulting in a detection rate of 81 %. Among the undetected events, 85 events went undetected due to low precipitation (not the cases of extreme precipitation), and 69 events went undetected due to a lack of GRACE data in certain months, resulting in the inability to obtain effective high-frequency signals. Among the remaining 309 undetected floods, the omission of 184 floods may have occurred due to the fact that the maximum daily precipitation was less than 50 mm, causing GRACE to fail to identify a flood signal. The other 125 undetected flood events may have been caused by GRACE itself failing to identify flood conditions. To view the spatial distributions of precipitation-type floods and the corresponding situation obtained from flood inundation data extracted from GRACE and MODIS, the reader is referred to the following

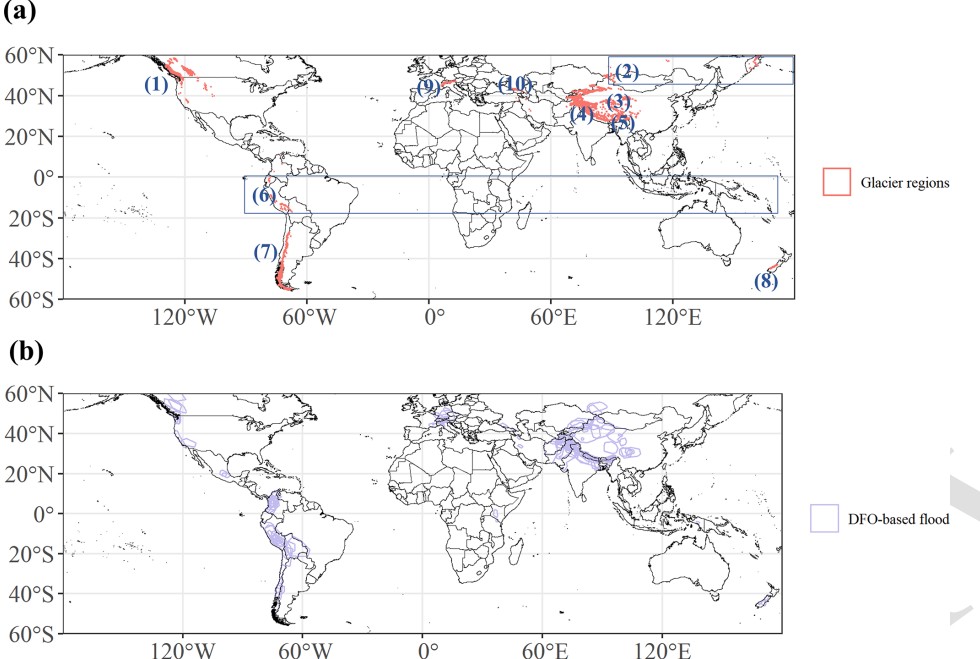

**Figure 5.** Global glacier distribution **(a)** and corresponding DFO-based flood events **(b)** for (1) glacial regions in western Canada and the US, (2) glacial regions in North Asia, (3) glacial regions in Central Asia, (4) glacial regions in the western South Asia, (5) glacial regions in the eastern South Asia, (6) glacial regions at low latitudes, (7) glacial regions in the southern Andes, (8) glacial regions in New Zealand, (9) glacial regions in central Europe and (10) glacial regions in central eastern Caucasus.

Zenodo repository: https://doi.org/10.5281/zenodo.6831105 (Zhang et al., 2022a). CE4

## 4.4 Further validation with news reports and social media

We further selected some flood events using news reports and social media (like Twitter and Weibo) to verify if there were some flood events that could not be identified by the DFO but could be identified by GRACE-based flood days. Figure 11 presents nine flood events not recorded by the DFO, including floods that occurred in different time periods and areas, such as the eastern US, northern and southern South America, Mozambique in Africa, France, India, China, Malaysia, Indonesia and Australia. The red boxes indicate the approximate location of the reported flood events. The blue areas were the GRACE-based flood days corresponding to the duration and approximate location of flood events recorded by social media or news reports. We found that GRACE-based flood days could identify these missing flood events well, which also proved the effectiveness of using GRACE to identify large-scale flood events. Our data can be used as a good supplement to DFO data.

## 4.5 Validation with discharge data

We also compared GRACE-based flood days with discharge data to assess our detection ability. We used the FloodR method of Fischer et al. (2021) to extract possible flood events from 3408 GRDC discharge data to serve as a basic reference standard when verifying the accuracy of the results extracted in this study. We focused on extreme precipitation-induced flood events and similarly constrained the results derived from the discharge data with extreme precipitation data. To ensure accuracy, we first selected the floods extracted from discharge stations with a GFS greater than 0.5 for comparison. Due to the fact that discharge reflects the amount of water integrated over its entire contributing basin and contributing time (Yang et al., 2019), we combined the flood events obtained from each discharge station in time series to describe the flood events in the 261 watersheds (HydroSHEDS Basins Level 4; Lehner and Grill, 2013; Lehner et al., 2008). Flood events in the same watershed were merged according to whether there was an intersection in the time series. The accuracy index used for comparison in this study was the probability of detection (POD) (Yang et al., 2021), i.e., whether each flood event in the river basin contained the GRACE-based flood days. Although flood events derived from discharge cannot guarantee that the surrounding land will experience flooding, they provide us with a reference to support the reliability of the time series verification process.

Figure 12 shows the global distribution of discharge locations and the GFS. The data distributions in North America, South America, Europe and southeastern Australia were relatively dense, whereas data were seriously missing in central,

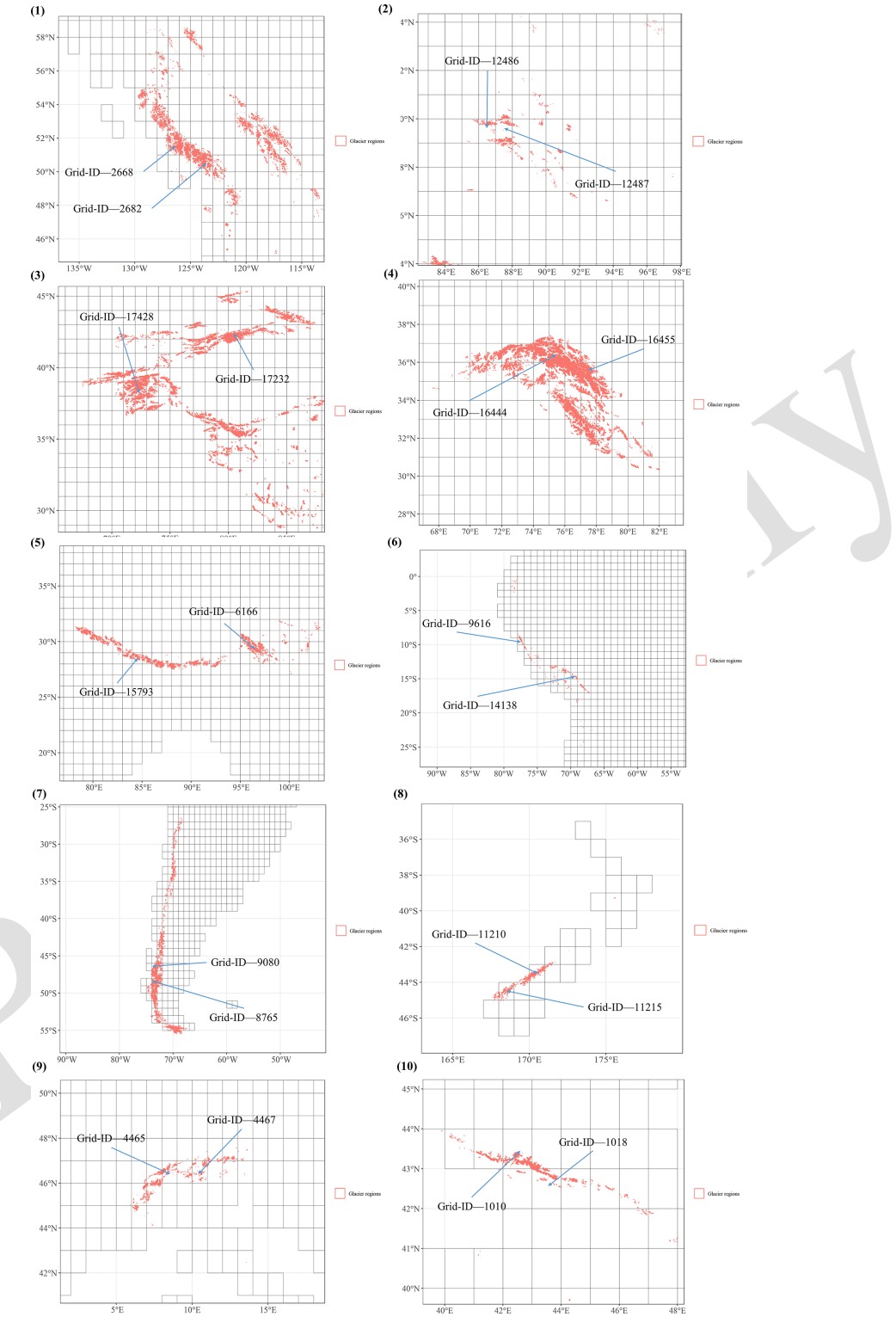

**Figure 6.** The 10 glacial regions within the 60° S–60° N latitudes. Two representative GRACE grid points in each region were selected to analyze the temporal detection of floods and correspond to the time series in Fig. 7. The 10 regions are as follows: (**1**) glacial regions in western Canada and the US, (**2**) glacial regions in North Asia, (**3**) glacial regions in Central Asia, (**4**) glacial regions in western South Asia, (**5**) glacial regions in eastern South Asia, (**6**) glacial regions at low latitudes, (**7**) glacial regions in the southern Andes, (**8**) glacial regions in New Zealand, (**9**) glacial regions in central Europe and (**10**) glacial regions in central eastern Caucasus.

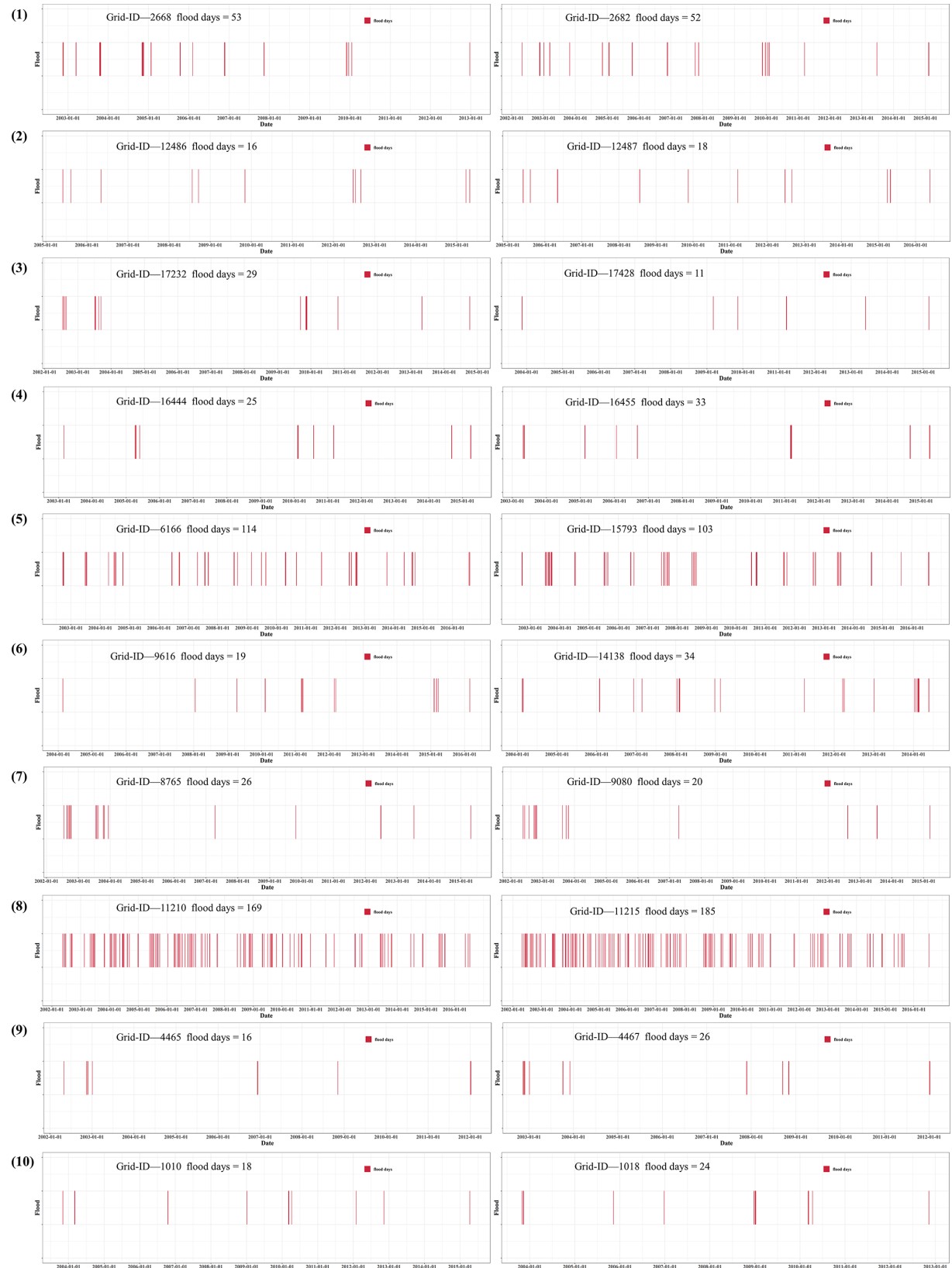

**Figure 7.** Flood detection results for different glacial regions (specific regions are consistent with Fig. 6).

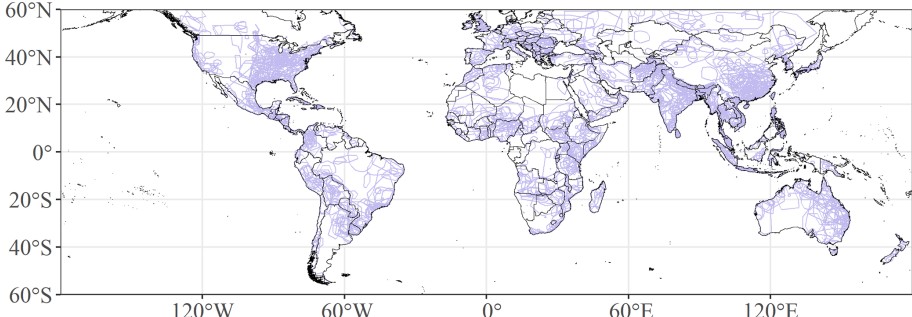

**Figure 8.** Spatial distribution of DFO-recorded flood events.

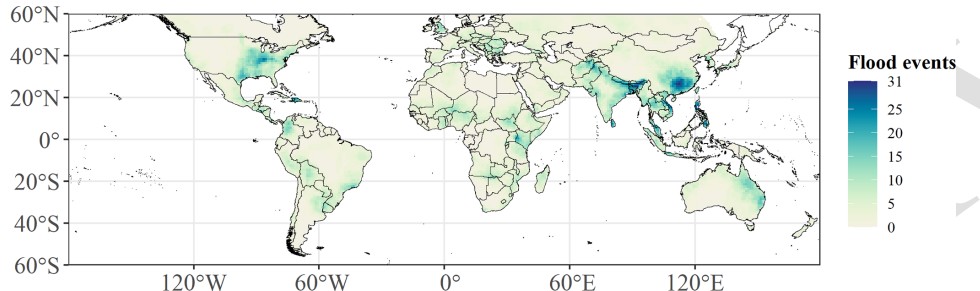

**Figure 9.** The frequency of flood events recorded by DFO distributed on 1° grids.

northern and eastern Asia. The areas with a higher GFS were located in the eastern US and central Europe, and the stations recorded in these areas were relatively complete. The discharge stations with a GFS above 0.5 accounted for 73.49 %
5 of stations. Figure 13 shows the flood events in the Level 4 river basins. We find that most flood events reflected by the discharge data were located in the eastern and western US, central South America, eastern Europe and New Zealand.

The POD calculation results are shown in Fig. 14: the
10 darker the color, the higher the corresponding flood detection accuracy. We found that the overall accuracy performed well; the detection accuracies obtained for the central and eastern parts of the US, western South America, southern Africa and around Australia were relatively high. Figure 15 shows the
15 histogram of 261 watersheds of Level 4 basins; the percentage of river basins with POD values greater than or equal to 0.5 is 62 %. This finding shows that our extracted flood days also reflected relatively high accuracies in comparison with flood events at river basins.
20    In the Table 1, we list the flood detection performance compared with DFO, MODIS and discharge, respectively. GRACE was able to detect 81 % of flood events recorded by DFO and 87 % of flood events recorded by MODIS. If we summed all flood events from the 261 river basins, GRACE-
25 based flood days could identify 53 % flood events derived from discharges. The percentage of river basins with POD values greater than or equal to 0.5 was 62 %.

## 4.6   Uncertainty analysis

The uncertainty analysis performed in this study mainly focused on the selection of the extreme precipitation threshold. The most common method for determining the extreme precipitation threshold is to use the quantile of the analyzed time series, considering either the quantile of the entire time series (QETS) data or the quantile of wet days with daily precipitation greater than 1 mm (QWDTS). This study compared the different PODs which were obtained by setting different quantile threshold scenarios when comparing with the DFO database. We selected the 90th, 95th and 99th quantiles for the two methods described above. Figure 16 shows that the selection of different thresholds in the two extreme-precipitation scenarios influenced the flood extraction accuracy of the POD, with contributions ranging from 72.4 % to 81.4 %. This shows that the selected thresholds can affect the detection rate of approximately 9 % (roughly 214) of the flood events. We also provide the six products derived based on these two constraints for further analysis and use by researchers.

## 5   Data usage instructions

The data obtained herein are provided using the polygon shapefile (SHP file) format. A separate file is provided for each day, and each file represents the global flood day distribution. The spatial resolution is 1°, covering the range of 60° S–60° N from 1 April 2002 to 31 August 2016. The SHP

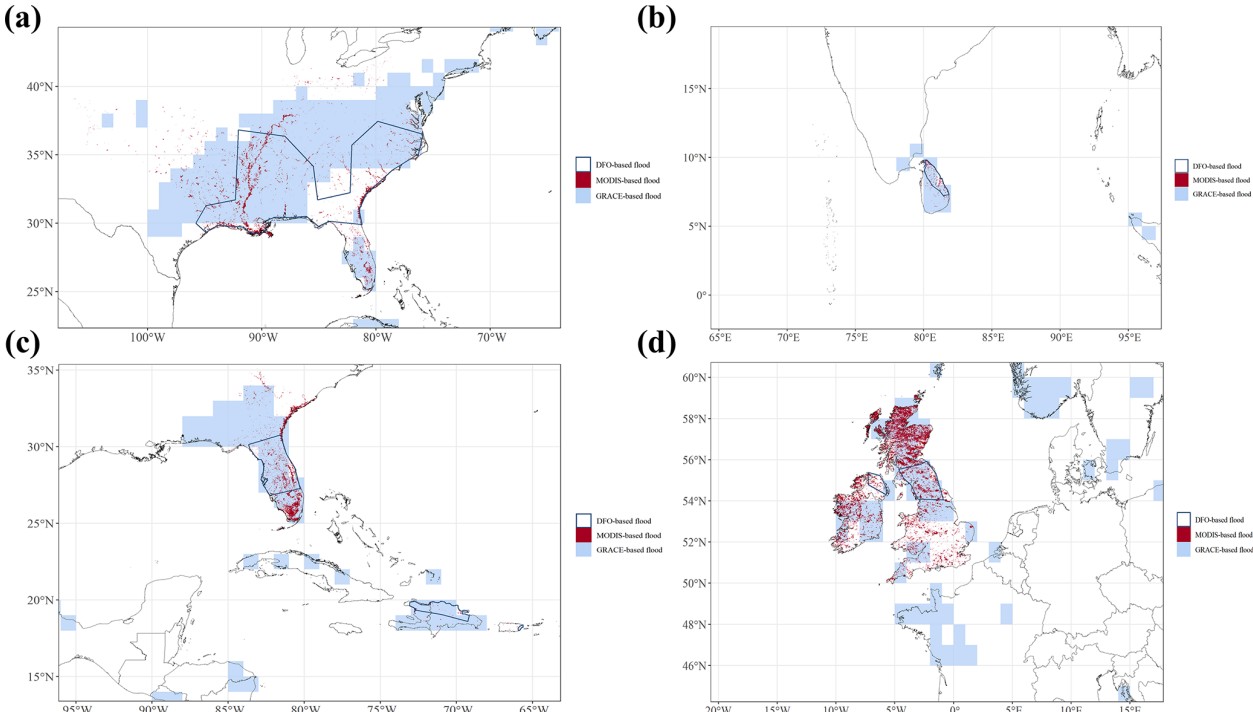

**Figure 10.** Flood inundation information recorded by the DFO (dark blue polygon), MODIS (red pixels) and GRACE (light blue) showing **(a)** the ID-2167 flood event that occurred from 22 February to 17 March 2003, **(b)** the ID-2601 flood event that occurred from 11 to 23 December 2004, **(c)** the ID-2566 flood event that occurred from 15 September to 1 October 2004 and **(d)** the ID-4319 flood event that occurred from 5 December 2015 to 26 January 2016.

**Table 1.** Flood detection performance compared with DFO, MODIS and discharge data.

|           | DFO (no. of flood events) | MODIS (no. of flood events) | Discharge (no. of flood events) | Discharge (no. of river basins) |
|-----------|---------------------------|------------------------------|----------------------------------|----------------------------------|
| Total     | 2380                      | 807                          | 10 472                           | 261                              |
| Detection | 1917                      | 703                          | 5597                             | 156 (POD $\geq$ 0.5)             |
| Percent   | 81 %                      | 87 %                         | 53 %                             | 62 %                             |

files have two fields, namely "ID" and "Value": ID represents the index number of the 1° grid, and Value is a binary variable (with a value of zero or one) indicating whether a flood occurred on a specific day. This flood day product can be used to analyze the spatial distributions of historical flood days within a 60° north–south latitude and to extract specific flood events in combination with historical data from observation sites. At the same time, the products are obtained based on observed data and can be used to verify flood model results. Considering that the El Niño–Southern Oscillation affects both drought and flood events in different parts of the world, these data can be used to further analyze the impacts of the El Niño–Southern Oscillation on flood days around the world.

## 6 Data availability

The flood day product produced in this study can be obtained from https://doi.org/10.5281/zenodo.6831384 (Zhang et al., 2022b).

## 7 Conclusion and discussion

This study successfully extracted global flood days using GRACE TWS and extreme precipitation data between 60° S and 60° N from 1 April 2002 to 31 August 2016. The results were compared in time and space with the flood events recorded by the DFO, MODIS and GRDC discharge data. It showed that GRACE-based flood events could identify 81 % of the flood events recorded by the DFO and 87 % flood events derived from MODIS. To further verify the reliability of our GRACE-based flood products, we compared them

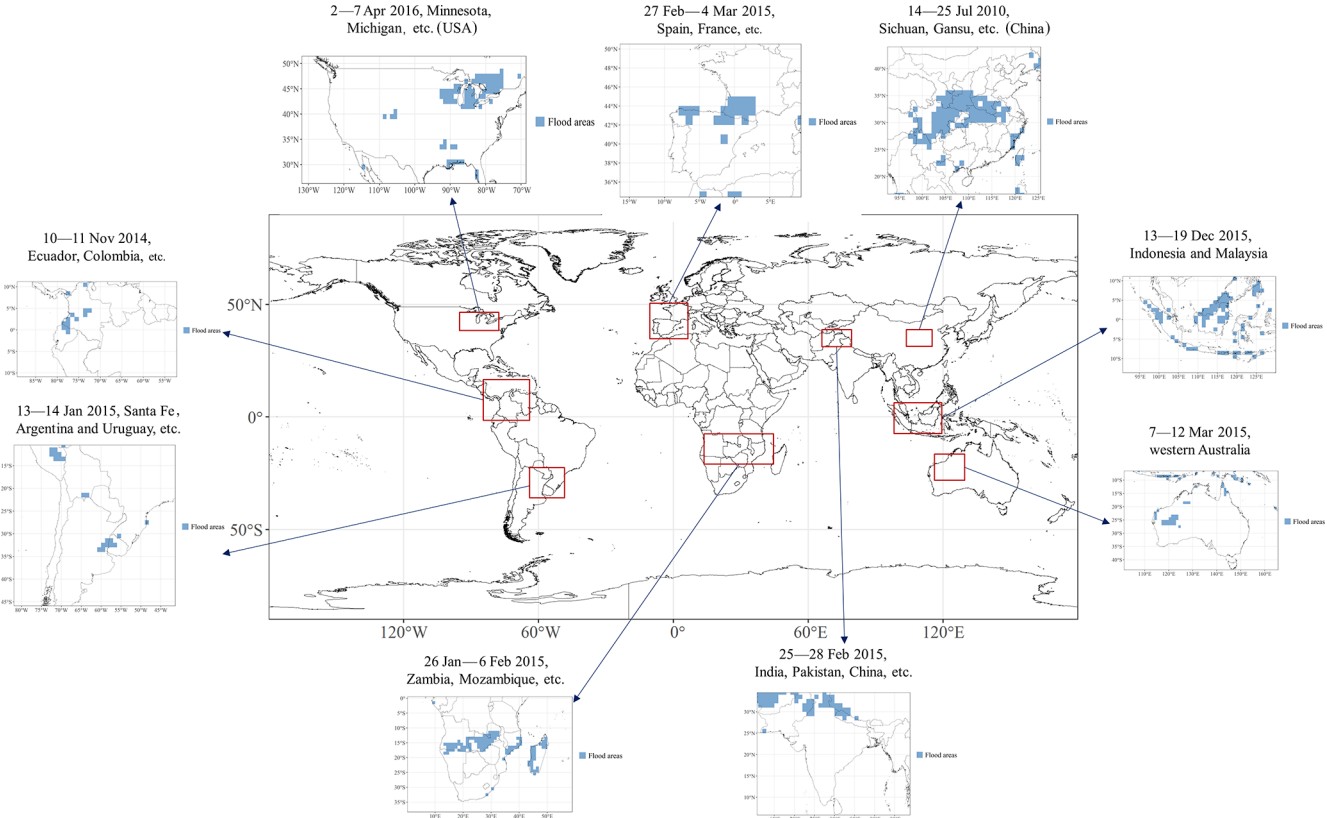

**Figure 11.** Validation of some flood events recorded by social media and news but not by the DFO.

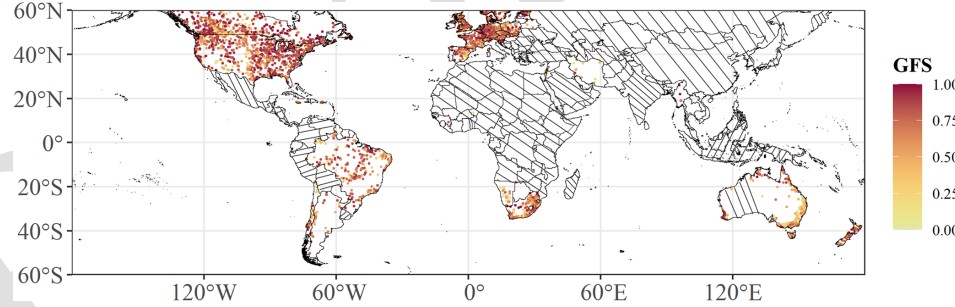

**Figure 12.** The GFS of 3408 discharge sites around the world. Hatching denotes no data.

with the flood events extracted from global GRDC discharge data, and the POD greater than or equal to 0.5 reached 62 % at the river basin scale. Moreover, we selected representative flood events not recorded by the DFO but recorded by social media or the news in different regions of the world as verification examples. These results also showed that our GRACE-based flood days could identify and supplement flood events not recorded by the DFO. The value of our product is mainly reflected in the following aspects. First, the GRACE-based flood days own wide coverage (covering between 60° S and 60° N). Second, the information is continuous in time and space, and the number of flood days in different areas or on different research timescales can be calculated according to

research needs, which makes up for the lack of flood events due to weather conditions in the MODIS dataset and missing records in the DFO dataset. Third, it provides not only important data support for the spatiotemporal distributions and attributions of global flood events but also a reference for large-scale quasi-real-time flood event monitoring with the development of GRACE-FO and the quality improvement of GRACE daily data.

However, we acknowledge that there are some limitations to these data. First, we used extreme precipitation to constrain the data, and the detection ability of some small floods was, thus, insufficient. Second, considering the regional differences in precipitation at the GRACE resolution level, the

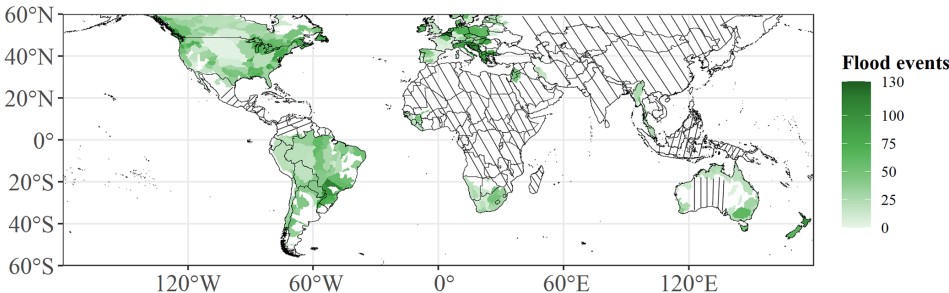

**Figure 13.** Flood event distribution in the Level 4 basins. Hatching denotes no data.

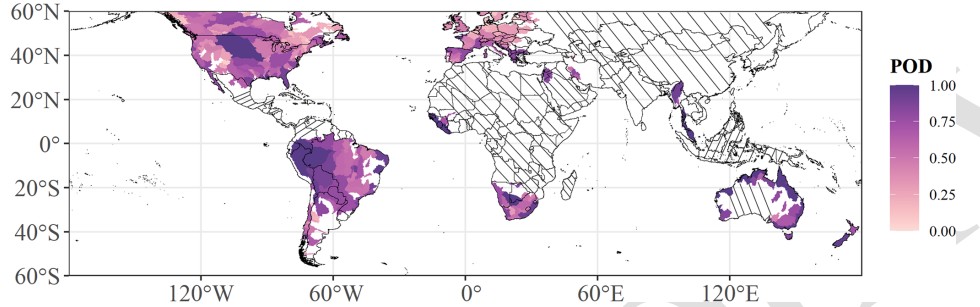

**Figure 14.** The POD values in the Level 4 basins. Hatching denotes no data.

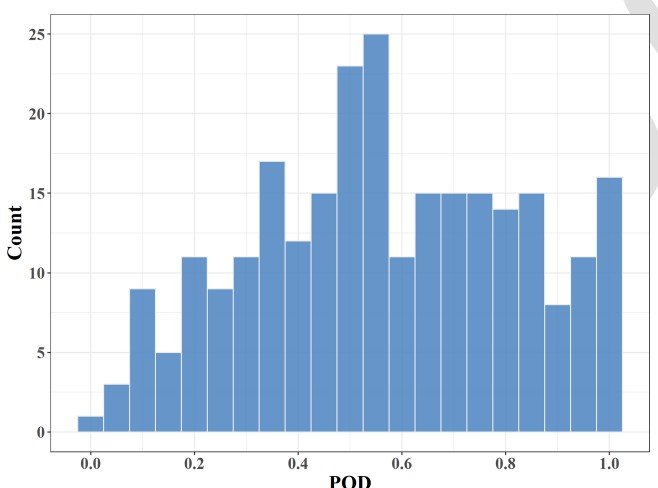

**Figure 15.** Histogram of the POD in the 261 river basins.

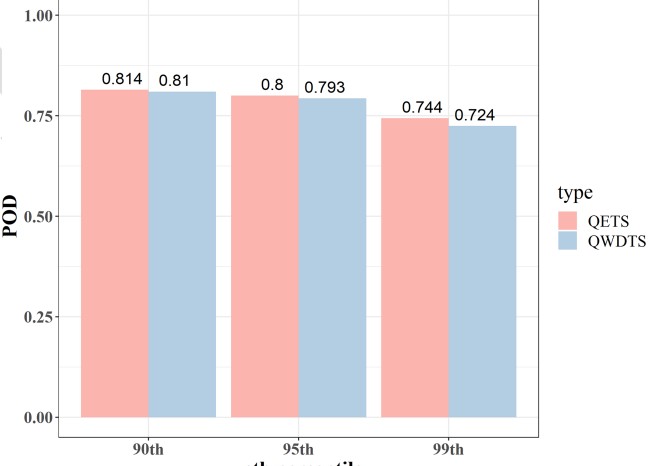

**Figure 16.** Influence of the selected threshold and extreme precipitation standard on the POD.

maximum precipitation under the GRACE grid can retain the signal of extreme precipitation to the greatest extent. We have also tried to take the mean value of the precipitation covered by the GRACE grid, but this led to many missing flood events. Third, the high-frequency signals of GRACE TWS may result in the loss of some flood events, as previously demonstrated. Although the FPI can supplement some flood events that were not identified by high-frequency signals, it can not guarantee that all flood events lost due to high-frequency signals could be accounted for. Fourth, the GRACE-based days are affected by ocean signals around

island countries due to the coarse data resolution, and researchers should be careful when using these data in such areas. Fifth, we were not able to compute the false detection of flood events. Due to observation difficulties, a complete and correct global record of floods is unavailable. This also highlights the importance of this study, which tries to provide a new approach for detecting global flood events. Although we could not calculate the false alarm rate, we could calculate the corresponding detection rate (i.e., POD) for the existing recorded floods and selected larger flood events recorded

by news reports or social media (not recorded by DFO) for further comparison. Sixth, we cannot correctly separate specific flood events from GRACE-based flood days nor can we separate false flood detection from unrecorded flood cases; these issues require further study in the future.

**Supplement.** The supplement related to this article is available online at: https://doi.org/10.5194/essd-15-1-2023-supplement.

**Author contributions.** KL, JZ and MW conceptualized the study, developed the methodology and carried out the investigation. JZ and KL undertook the validation and formal analysis. JZ procured resources, was responsible for data curation, prepared the manuscript and created the figures. KL reviewed and edited the manuscript, supervised the study, was responsible for the project administration and acquired funding. All authors have read and agreed upon the published version of the paper.

**Competing interests.** The contact author has declared that none of the authors has any competing interests.

**Disclaimer.** Publisher's note: Copernicus Publications remains neutral with regard to jurisdictional claims in published maps and institutional affiliations.

**Acknowledgements.** The research in this article was supported by the National Natural Science Foundation of China (grant no. 41771538). The financial support is highly appreciated. We are also grateful for data support from the GRACE daily solution, Institute of Geodesy, Graz University of Technology.

**Financial support.** This research has been supported by the National Natural Science Foundation of China (grant no. 41771538).

**Review statement.** This paper was edited by Alexander Gelfan and reviewed by four anonymous referees.

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

## Remarks from the language copy-editor

CE1    Please confirm the slight change.
CE2    Please confirm the change.
CE3    Please confirm that the sentence is now correct.
CE4    Please confirm the change.

## Remarks from the typesetter

TS1    Please confirm change.
TS2    Please give an explanation of why this needs to be changed. We have to ask the handling editor for approval. Thanks.
TS3    Please give an explanation of why this needs to be changed. We have to ask the handling editor for approval. Thanks.
TS4    Please give an explanation of why this needs to be changed. We have to ask the handling editor for approval. Thanks.