# Peer review of "Flood Detection Using GRACE Terrestrial Water Storage and Extreme Precipitation"

_Earth System Science Data, 2022_

## Referee Comment (RC1)

The paper and related dataset are interesting and present significant effort in evaluating and monitoring flood events. However, the methods description and error assessment of the dataset are somewhere too vague and uncertain. The main concerns regarding various parts of the manuscript are presented below.

1) GPM, that is mentioned as the single source material for precipitation (section 2.2), has been operational since 2014. Also, precipitation series are necessary for the flood dataset construction (figure 1). However, obtained flood dataset is cover period from 2002 to 2016. It remains unclear how the dataset was obtained for 2002-2014.

2) As it was mentioned in the manuscript (section 2.1), the research used GRACE dataset that is based on set of spherical harmonic coefficients up to 40 degree and power. That is roughly equal to spatial resolution 20000/40=500 км or 5°. At the same time, GPM spatial resolution is 0.1°. In manuscript it was mentioned "we take the maximum values of the precipitation data under the GRACE grid coverage to further calculate the flood potential index and the number of extreme precipitation days". Such difference in spatial resolution between the datasets makes me wonder if the maximum is the best metric in this case.

3) The main disadvantage of this manuscript, which does not allow a full and clear assessment of the resulting data set, is absence of false alarm ratio or some other similar metric to understand how often received dataset falsely detect flood event.

---

## Author Comment (AC2)

We would like to thank the reviewers for the thorough reading of the manuscript and the valuable remarks that helped us to improve the manuscript. We have revised the manuscript carefully according to the reviewer's comments, and have incorporated the suggestions into the revised manuscript.

The notes below provide a point-by-point response to each comment from the referees. The texts with blue font are the reviewer's original comments, the texts with black font are authors' responses.

Major Commentsï¼

1. How do you consider the detected flood (extreme values in TWSA) over the glacier regions? I do not see the demonstration for these regions.

**Response**: In this study, we detect flood events by identifying change in high frequency signals of GRACE. The high-frequency signals capture the impact of extreme precipitation which has high magnitude and relatively short duration. While the melting of glaciers is usually a slower process and cannot be identified by high frequency signals. Therefore, in this paper, the floods we extracted are mainly precipitation-induced floods. Glacier-melting-induced floods are not considered in this study. To avoid misunderstanding, we have modified the title of this paper as follows:

*Precipitation-induced Flood Detection Using GRACE Terrestrial Water Storage*

2. Which term does the "high frequency signal" with the STL decomposition for the GRACE TWS? Does the final signal include a seasonal signal? It is not clear in Section 3.1, Please provide detailed explanations. I think sometimes the seasonal signal can also relate to floods.

**Response**: We thank reviewer's suggestions. The "high frequency signal" means the "remainder" in the STL decomposition. The "remainder" is the remaining part of the original data after the STL decomposition method which eliminates the seasonal and trend components. The TWS seasonal signal is also associated with flood mainly when the peak of the seasonal time series encounters heavy precipitation, which may trigger flood. Since the impact signal exerted by heavy precipitation is already included in the high-frequency signals of TWS, the above situation can also be detected.

3. How do you extract the flood day for different cases, I think the authors should show some figures (maybe time series) for a detailed demonstration. I do not think Section 3.4 is clear enough.

**Response**: We thank reviewer's suggestions. Figure 1 shows the intermediate process of flood detection for randomly selected grid. The steps are as follows. Firstly, we extracted the high-frequency signal of TWS using the STL method; Secondly, we calculated the possible flood days using the GESD method; Thirdly, we use flood potential index to supplement possible flood days in case the daily GRACE TWS data have lost useful high-frequency signals due to the interpolation process; Finally, we constrained the pre-selected floods using the extreme precipitation days derived from daily and cumulative precipitation. To explain the method more clearly, we will add a more detailed additional description of the flood days extraction process in the revised manuscript.

[Figure]

Figure 1 Schematic diagram of the intermediate variable time series processing for flood extraction based on GRACE and extreme precipitation for a randomly selected grid from Apr. 1st, 2002 to Aug. 31st,

2016. (a) GRACE TWS times series; (b) high-frequency signals time series derived from GRACE and flood days extraction based on GESD; (c) flood days extraction based on flood potential index time series; (d) daily precipitation time series and extreme precipitation days based on 95-th quantile; (e) 3-day cumulative precipitation time series and extreme precipitation days based on 95-th quantile; (f) 5-day cumulative precipitation time series and extreme precipitation days based on 95-th quantile.

**4. Which version of ITSG was used in this study? Please list it in Section 2.1.**

**Response**: We thank reviewer's suggestions. We chose the GRACE daily solution based on ITSG-Grace2018 gravity field model. The ITSG-Grace2018 gravity field model, which offers unconstrained monthly and Kalman-smoothed daily solutions, is the most recent GRACE-only gravity field model computed in Graz. It is a reprocessing of the whole GRACE time series starting from 2002-04. We will add this in the revised manuscript.

**5. Section 3.2, how do you evaluate/consider the reliability of the GESD test, please give a more detailed demonstration.**

**Response**: We thank reviewer's suggestions. GESD test is a commonly used univariate anomaly detection algorithm. It has been widely used in the field of hydrological anomaly detection (Saghafian et al., 2014; Clark and Zipper, 2016).GESD test is mainly used in this study to extract possible flood days corresponding to the high-frequency signals. In this study, the method to be selected for extracting flood information from the high-frequency signals should ensure that it was affected by the random error in the high-frequency signal as little as possible and that the flood signals were extracted as much as possible. The GESD test method overcomes the primary limitation of the Grubbs test and the Tietjen-Moore test that the suspected number of outliers, k, must be specified exactly. The GESD test only requires that an upper bound for the suspected number of outliers be specified (Rosner, 1983). These advantages of the GESD test method were highly compatible with our needs.

Moreover, in the practical operation for our dataset, the method performed well. On the one hand, the method effectively extracted the important anomalous peaks (i.e., the pre-extracted flood days), and the extracted flood days for each spatial grid were stable and did not increase with the increase of the preset up bound. As shown in Fig. 2, the histogram distribution of the pre-extracted global flood days for each spatial grids was concentrated around 200 days, not more than 500 days. It was not more than 200 flood events after converted from flood days to flood events. The order of magnitude was consistent with the number of historical flood events. Finally, the pre-extracted flood based on GESD test was an important part of the final extracted flood days, and the

good validation of the final product also confirmed that GESD test method played a reliable and important role in the intermediate processing. We will add a more detailed description of the GESD test method in the revised manuscript.

[Figure]

Figure 2 The histogram of pre-selected flood days based on high-frequency signals of TWS using GESD test method.

6. When I check the supplementary materials, I find that there are many differences among the spatial patterns of DFO-based, MODIS-based, and GRACE-based floods. More discussion should be drawn to improve this MS, the work in Gouweleeuw (2018) also showed that ITSG might not catch the flooded area well. Besides, how do you consider the different representation between terrestrial water storage (TWS) and runoff for one flood event?

**Response**: We thank reviewer's suggestions. The DFO database mainly records large-scale flood events from news and government announcements and misses some flood events. Although DFO records the start and end times and the approximate spatial locations of flood events, the temporal range is sometimes long (more than one or two months), and the spatial locations are only roughly delineated according to news reports, which contained large uncertainties in spatial location and duration. Moreover, although flood detection methods based on MODIS data could finely delineated the spatial inundation ranges, it was mainly aimed at specific flood events in small areas. Only a limited amount of flooding can be identified by remote sensing images due to the influence of bad weather. MODIS resolution is 1 km, which is a higher resolution product compared to GRACE TWS (1°, ~100km). The flood products we provided can only indicate the presence of flood events under the ~100km grid coverage, and specific detailed locations require further identification by high-resolution satellite remote sensing image. Therefore, we can see differences among the spatial patterns of the three datasets. However, as we show in Figure 8-9 and supplementary figures, there is still an overwhelming majority of results indicating a consistent spatial pattern of flood

inundation. Our products can provide a sufficient data foundation for large-scale research on the spatiotemporal distributions and attributions of global floods. We will further deepen the discussion and analysis of this result in the subsequent revision.

As for the different representation between terrestrial water storage (TWS) and runoff for one flood event, we extracted the flood days based on GRACE. To facilitate the comparison between GRACE-based floods and discharge-based floods, we first extracted the corresponding flood events with the discharge data using the FloodR method, and then compared whether each flood event period contained the flood days we extracted. This was used to calculate the effectiveness of our product in comparing to floods derived from discharge.

Minor Comments:

1. Figure 3, the tick values of the x-axis should be denser.

**Response**: We thank reviewer's suggestions. We have made the x-axis denser.

2. P2L69ï¼ what is the "other places"?

**Response**: We thank reviewer's suggestions. We have changed the "other" to a more critical description, which reads as follows.

*The numbers of flood events recorded in China, Russia and Canada are obviously lacking.*

3. "Diksha Gupta et al (Gupta and Dhanya, 2020)"… in all the MS, these citation formats are strange, maybe wrong.

**Response**: We thank reviewer's suggestions. We have modified it to the correct format

4. P3L109, what is the difference between the two citations of "Dill, 2008; Dill et al., 2008"?

**Response**: We thank reviewer's suggestions. One was misquoted here and has been removed.

5. P10L296, "Tellman1" to "Tellman"

**Response**: We thank reviewer's suggestions. We have modified it.

6. The two references share the same data name, I think there is no need to put them on two websites. In addition, there is a mistake in the name of the authors.

"Zhang, J., Liu, K., and Wang, M.: Flood detection using GRACE Terrestrial Water Storage and Extreme Precipitation (1.0.0). Zenodo. https://doi.org/10.5281/zenodo.6831105, 2022a.

Zhang, j., Liu, K., and Wang, M.: Flood Detection Using GRACE Terrestrial Water Storage and Extreme Precipitation (1.0.0) [Data set]. Zenodo. https://doi.org/10.5281/zenodo.6831384, 2022b."

**Response**: We thank reviewer's suggestions. Since each uploaded file generates a unique DOI, we couldn't put them on one website. We have modified the mistake in the name of the authors.

7. I do not think the reference format is right. "Mayer-Gürr, T., Behzadpour, S., Kvas, A., Ellmer, M., Klinger, B., Strasser, S., and Zehentner, N.: ITSG-Grace2018: monthly, daily and static gravity field solutions from GRACE, 2018."

**Response**: We thank reviewer's suggestions. We have modified this reference format in subsequent revisions.

8. The references should be checked carefully.

**Response**: We thank reviewer's suggestions. We have checked all citation formats of this paper.

**References:**

Clark, Elyse V and Zipper, Carl E: Vegetation influences near-surface hydrological characteristics on a surface coal mine in eastern USA, Catena, 139, 241-249, 2016.

Rosner, Bernard: Percentage points for a generalized ESD many-outlier procedure, Technometrics, 25, 165-172, 1983.

Saghafian, Bahram, Golian, Saeed, and Ghasemi, Alireza: Flood frequency analysis based on simulated peak discharges, Natural Hazards, 71, 403-417, 2014.

---

## Author Comment (AC3)

We would like to thank the reviewers for the thorough reading of the manuscript and the valuable remarks that helped us to improve the manuscript. We have revised the manuscript carefully according to the reviewer's comments, and have incorporated the suggestions into the revised manuscript.

The notes below provide a point-by-point response to each comment from the referees. The texts with blue font are the reviewer's original comments, the texts with black font are authors' responses.

1. From the figures in the MS, the mountain glacier regions are included. I do not mean the glacier-melting-induced floods. I do not see the results in these regions, the detection of floods in these regions should be shown to reviewers.

**Response**:We thank reviewer's suggestions. We used global glacier outline data from the Randolph Glacier Inventory (RGI). This dataset can be used to estimate glacier volumes, rates of elevation change at regional and global scales, and the response of the cryosphere to climate forcing. The dataset is updated annually in shapefile format. We used it in this paper to locate global glaciers (Arendt et al., 2017).

In the range of 60°S—60°N, there are 10 glacier regions and 163 flood events recorded based on DFO database (Figure 1). 142 flood events were identified and 21 flood events were not detected, with a POD of 0.87. The capacity of flood detection is close to the global POD (0.81). The results showed that GRACE also has good potential in identifying precipitation-induced floods in glaciers regions. Of these 21 flood events, 4 flood events could not be identified due to missing months in GRACE data. 8 flood events had a maximum daily precipitation of less than 40 mm according to the DFO-recorded time period and spatial location (minimum: 8.44 mm, maximum: 36.56 mm) and GRACE could not identify the weaker signal. The remaining 9 flood events could not be identified due to GRACE itself failing to identify flood conditions.

We further selected the GRACE grid covering the glacier regions and analyzed the characteristics of the extracted flood days. Figure 2 showed the 10 detailed glacier regions and the corresponding selected GRACE grids which covered the main glacier areas. Figure 3 showed the results of the extracted flood days related to the grid of each region. In general, the number of flood days in the glacier regions was relatively small, and mostly concentrated within 50 days from Apr. 1st, 2002 to Aug 31st, 2016, while the glaciers in the South Island of New Zealand and the glacier regions in the east of southern Asia were exceeded 100 days. The South Island of New Zealand where mountain glaciers are located experiences a hyper-maritime climate, and the west coast of the South Island receives the most precipitation, with annual precipitation >12 m (Anderson et al., 2010). Glacier regions in the east of south Asia are mainly located in the Himalayas, where normal climatic fluctuations become rather quick in the Himalayan sectors due to topography and the southwest Indian Ocean monsoon. It would occur cloud bursts, high winds, snowstorms, etc., further caused quick floods

(Nandargi and Dhar, 2011).

**(a)**

[Figure]

**(b)**

Figure 1 Global glacier distribution (a) and corresponding DFO-based flood events (b). (1) glacier regions in the western Canada and US; (2) glacier regions in north Asia; (3) glacier regions in central Asia; (4) glacier regions in the west of south Asia; (5) glacier regions in the east of south Asia; (6) glacier regions in the low latitudes; (7) glacier regions in the southern Andes; (8) glacier regions in the New Zealand; (9) glacier regions in the central Europe; (10) glacier regions in the middle east of Caucasus.

[Figure]

Figure 2 Ten glacier regions during the 60°S—60°N latitudes. two representative GRACE grid points in each region were selected to analyze the flood of the temporal detection. This was corresponding to the time series in Figure 3. (1) glacier regions in the western Canada and US; (2) glacier regions in north Asia; (3) glacier regions in central Asia; (4) glacier regions in the west of south Asia; (5) glacier regions in the east of south Asia; (6) glacier regions in the low latitudes; (7) glacier regions in the southern Andes; (8) glacier regions in the New Zealand; (9) glacier regions in the central Europe; (10) glacier regions in the middle east of Caucasus.

[Figure]

Figure 3 Flood detection results for different glacier regions.(specific regions are consistent with Figure-2.)

2. In Figure 1 in your response, you should also show the location of the grid.

**Response**: We thank reviewer's suggestions. We have shown the location of the grid.

[Figure]

Figure 4 Schematic diagram of the intermediate variable time series processing for flood extraction based on GRACE and extreme precipitation for a randomly selected grid from Apr. 1st, 2002 to Aug. 31st, 2016. (a) randomly selected grid for flood detection; (b) GRACE TWS times series; (c) high-frequency signals time series derived from GRACE and flood days extraction based on GESD; (d) flood days extraction based on flood potential index time series; (e) daily precipitation time series and extreme precipitation days based on 95-th quantile; (f) 3-day cumulative precipitation time series and extreme precipitation days based on 95-th quantile; (g) 5-day cumulative precipitation time series and extreme precipitation days based on 95-th quantile.

**References:**

Anderson, Brian, Mackintosh, Andrew, Stumm, Dorothea, George, Laurel, Kerr, Tim, Winter-Billington, Alexandra, and Fitzsimons, Sean: Climate sensitivity of a high-precipitation glacier in New Zealand, Journal of Glaciology, 56, 114-128, 2010.

Arendt, A, Bliss, A, Bolch, T, Cogley, JG, Gardner, A, Hagen, J-O, Hock, R, Huss, M, Kaser, G, and Kienholz, C: Randolph Glacier inventory–A dataset of Global glacier outlines: Version 6.0: Technical report, Global land ice measurements from space, 2017.

Nandargi, S and Dhar, ON: Extreme rainfall events over the Himalayas between 1871 and 2007, Hydrological Sciences Journal, 56, 930-945, 2011.

---

## Author Response (AR1)

We would like to thank the reviewers for the thorough reading of the manuscript and the valuable remarks that helped us to improve the manuscript. We have revised the manuscript carefully according to the reviewer's comments, and have incorporated the suggestions into the revised manuscript.

The notes below provide a point-by-point response to each comment from the referees. The texts with blue font are the reviewer's original comments, the texts with black font are authors' responses.

**Response to reviewer #1**

The paper and related dataset are interesting and present significant effort in evaluating and monitoring flood events. However, the methods description and error assessment of the dataset are somewhere too vague and uncertain. The main concerns regarding various parts of the manuscript are presented below.

1. GPM, that is mentioned as the single source material for precipitation (section 2.2), has been operational since 2014. Also, precipitation series are necessary for the flood dataset construction (figure 1). However, obtained flood dataset is cover period from 2002 to 2016. It remains unclear how the dataset was obtained for 2002-2014.

**Response**:We thank reviewer's suggestions. The precipitation data we acquired comes from NASA's Global Precipitation Measurement IMERG Final Run product. The IMERG dataset now includes TRMM-era data going back to June 2000. We can find the detailed descriptions at the website https://gpm.nasa.gov/data/directory.

2. As it was mentioned in the manuscript (section 2.1), the research used GRACE dataset that is based on set of spherical harmonic coefficients up to 40 degree and power. That is roughly equal to spatial resolution 20000/40=500 ÐºÐ¼ or 5°. At the same time, GPM spatial resolution is 0.1°. In manuscript it was mentioned "we take the maximum values of the precipitation data under the GRACE grid coverage to further calculate the flood potential index and the number of extreme precipitation days". Such difference in spatial resolution between the datasets makes me wonder if the maximum is the best metric in this case.

**Response**:We thank reviewer's suggestions. There is indeed a difference in resolution between GRACE and GPM. We focus on flood events caused by extreme precipitation, so effectively extracting extreme precipitation signals is the basis for ensuring the integrity of flood event extraction. Considering the regional differences of precipitation at GRACE resolution level, the maximum precipitation under the GRACE grid can retain the signal of extreme precipitation to the greatest extent. We have also tried to take the mean value of the precipitation corresponding to the GRACE grid coverage,

but this leads to a missing of many flood events. We have added this discussion in the Discussion section in Lines 499-502, Page 21.

*Second, considering the regional differences of precipitation at GRACE resolution level, the maximum precipitation under the GRACE grid can retain the signal of extreme precipitation to the greatest extent. We have also tried to take the mean value of the precipitation covered by GRACE grid, but this led to many missing flood events.*

3. The main disadvantage of this manuscript, which does not allow a full and clear assessment of the resulting data set, is absence of false alarm ratio or some other similar metric to understand how often received dataset falsely detect flood events.

**Response**:We thank reviewer's suggestions.

We agree reviewer's suggestions that false detection of flood events is important. However, due to the difficulty in observation, a complete global record of floods is unavailable. This also highlights the importance of this study which tries to provide a new approach for detecting global flood events. The most available reference observation datasets include DFO and GRDC discharge datasets. The number of flood events recorded by the DFO data is also incomplete, the global distribution of GRDC discharge data is uneven and some measurements are still missing in time series. Moreover, we couldn't correctly separate specific flood events from GRACE-based flood days, which is also direction that needs further study in the future. We therefore cannot calculate the false alarm rate based on the available incomplete dataset. However, we calculated the probability of detection (POD) of DFO (including MODIS-based flood inundation) and GRDC discharge based on existing observational datasets, and selected larger flood events recorded by news media (not recorded by DFO) for further comparison. Although we cannot calculate the false alarm rate, we can calculate the corresponding detection rate (i.e. POD) for the existing recorded floods. We also acknowledged that we could not effectively separate false flood detection and non-recorded flood cases. We have discussed these shortcomings in Discussion section in Lines 506-510, Page 21.

*Fifth, we were not able to compute the false detection of flood events. Due to the difficulty in observation, a complete and correct global record of floods is unavailable. This also highlights the importance of this study which tries to provide a new approach for detecting global flood events. Although we cannot calculate the false alarm rate, we can calculate the corresponding detection rate (i.e. POD) for the existing recorded floods and selected larger flood events recorded by news reports or social medias (not recorded by DFO) for further comparison. Sixth, we cannot correctly separate specific flood events from GRACE-based flood days and separate false flood detection from non-recorded flood cases, which are also direction that needs further study in the future.*

**Response to reviewer #2**

**Major Commentsï¼**

1. How do you consider the detected flood (extreme values in TWSA) over the glacier regions? I do not see the demonstration for these regions.

Second comments:    From the figures in the MS, the mountain glacier regions are included. I do not mean the glacier-melting-induced floods. I do not see the results in these regions, the detection of floods in these regions should be shown to reviewers.

**Response**:We thank reviewer's suggestions. We have shown and demonstrated the detection of floods in the mountain glacier regions in Lines 313-350, Pages 11-14

*We used global glacier outline data from the Randolph Glacier Inventory (RGI). This dataset can be used to estimate glacier volumes, rates of elevation change at regional and global scales, and the response of the cryosphere to climate forcing. The dataset is updated annually in shapefile format. We used it in this paper to locate global glaciers (Arendt et al., 2017).*

*In the range of 60°S—60°N, there are 10 glacier regions and 163 flood events recorded based on DFO database (Figure 1). 142 flood events were identified and 21 flood events were not detected, with a POD of 0.87. The capacity of flood detection is close to the global POD (0.81). The results showed that GRACE also has good potential in identifying precipitation-induced floods in glaciers regions. Of these 21 flood events, 4 flood events could not be identified due to missing months in GRACE data. 8 flood events had a maximum daily precipitation of less than 40 mm according to the DFO-recorded time period and spatial location (minimum: 8.44 mm, maximum: 36.56 mm) and GRACE could not identify the weaker signal. The remaining 9 flood events could not be identified due to GRACE itself failing to identify flood conditions.*

*We further selected the GRACE grid covering the glacier regions and analyzed the characteristics of the extracted flood days. Figure 2 showed the 10 detailed glacier regions and the corresponding selected GRACE grids which covered the main glacier areas. Figure 3 showed the results of the extracted flood days related to the grid of each region. In general, the number of flood days in the glacier regions was relatively small, and mostly concentrated within 50 days from Apr. 1st, 2002 to Aug 31st, 2016, while the glaciers in the South Island of New Zealand and the glacier regions in the east of southern Asia were exceeded 100 days. The South Island of New Zealand where mountain glaciers are located experiences a hyper-maritime climate, and the west coast*

*of the South Island receives the most precipitation, with annual precipitation >12 m (Anderson et al., 2010). Glacier regions in the east of south Asia are mainly located in the Himalayas, where normal climatic fluctuations become rather quick in the Himalayan sectors due to topography and the southwest Indian Ocean monsoon. It would occur cloud bursts, high winds, snowstorms, etc., further caused quick floods (Nandargi and Dhar, 2011).*

[Figure]

Figure 1 Global glacier distribution (a) and corresponding DFO-based flood events (b). (1) glacier regions in the western Canada and US; (2) glacier regions in north Asia; (3) glacier regions in central Asia; (4) glacier regions in the west of south Asia; (5) glacier regions in the east of south Asia; (6) glacier regions in the low latitudes; (7) glacier regions in the southern Andes; (8) glacier regions in the New Zealand; (9) glacier regions in the central Europe; (10) glacier regions in the middle east of Caucasus.

[Figure]

Figure 2 Ten glacier regions during the 60°S—60°N latitudes. two representative GRACE grid points in each region were selected to analyze the flood of the temporal detection. This was corresponding to the time series in Figure 3. (1) glacier regions in the western Canada and US; (2) glacier regions in north Asia; (3) glacier regions in central Asia; (4) glacier regions in the west of south Asia; (5) glacier regions in the east of south Asia; (6) glacier regions in the low latitudes; (7) glacier regions in the southern Andes; (8) glacier regions in the New Zealand; (9) glacier regions in the central Europe; (10) glacier regions in the middle east of Caucasus.

[Figure]

Figure 3 Flood detection results for different glacier regions.(specific regions are consistent with Figure 2.)

2. Which term does the "high frequency signal" with the STL decomposition for the GRACE TWS? Does the final signal include a seasonal signal? It is not clear in Section 3.1, Please provide detailed explanations. I think sometimes the seasonal signal can also relate to floods.

**Response**: We thank reviewer's suggestions. The "high frequency signal" means the "remainder" in the STL decomposition. The "remainder" is the remaining part of the original data after the STL decomposition method which eliminates the seasonal and trend components. The TWS seasonal signal is also associated with flood mainly when the peak of the seasonal time series encounters heavy precipitation, which may trigger flood. Since the impact signal exerted by heavy precipitation is already included in the high-frequency signals of TWS, the above situation can also be detected. We have provided detailed explanations in Lines 165-170, Page 5.

*Seasonal and trend decomposition using loess (STL) (Robert et al., 1990) is a filtering process as well as a general and robust time series decomposition and forecasting method used to decompose time series variables into seasonal, trend and remainder components for further forecasting. This process can handle data with any type of seasonality as well as high-frequency signal data. It also allows seasonal components to vary over time and is robust to outliers. In this study, we selected this method as a high-pass filtering tool to process GRACE TWS and obtain high-frequency signals (excluding seasonal and trend components) for subsequent analyses.*

3. How do you extract the flood day for different cases, I think the authors should show some figures (maybe time series) for a detailed demonstration. I do not think Section 3.4 is clear enough.

Second comments: In Figure 1 in your response, you should also show the location of the grid.

**Response**: We thank reviewer's suggestions. To explain the method more clearly, we have shown some figures for a detailed demonstration on the flood days extraction process in Lines 237-243, Page 7.

*In order to present the calculation process more clearly, we randomly selected a spatial grid for detailed processing. Figure S4 showed the intermediate process of flood days extraction. 1) we firstly extracted the high-frequency signal of TWS using the STL method; 2) then we calculated the possible flood days using the GESD method; 3) Next, we use flood potential index to supplement possible flood days in case the daily GRACE TWS data have lost useful high-frequency signals due to the interpolation process. 4) Finally we constrained the pre-selected floods using the extreme precipitation days*

*derived from daily and cumulative precipitation.*

[Figure]

Figure S4 Schematic diagram of the intermediate variable time series processing for flood extraction based on GRACE and extreme precipitation for a randomly selected grid from Apr. 1st, 2002 to Aug. 31st, 2016. (a) randomly selected grid for flood detection; (b) GRACE TWS times series; (c) high-frequency signals time series derived from GRACE and flood days extraction based on GESD; (d) flood days extraction based on flood potential index time series; (e) daily precipitation time series and extreme precipitation days based on 95-th quantile; (f) 3-day cumulative precipitation time series and extreme precipitation days based on 95-th quantile; (g) 5-day cumulative precipitation time series and extreme precipitation days based on 95-th quantile.

4. Which version of ITSG was used in this study? Please list it in Section 2.1.

**Response**: We thank reviewer's suggestions. We chose the GRACE daily solution based on ITSG-Grace2018 gravity field model. The ITSG-Grace2018 gravity field model, which offers unconstrained monthly and Kalman-smoothed daily solutions, is the most recent GRACE-only gravity field model computed in Graz. It is a reprocessing of the whole GRACE time series starting from 2002-04. We have added it in Lines 102-107, Page 3.

*The daily GRACE data selected in this study come from daily solutions obtained using Kalman smoothing by Mayer-Gürr et al of Graz University of Technology based on ITSG-Grace2018 gravity field model. The ITSG-Grace2018 gravity field model, which offers unconstrained monthly and Kalman-smoothed daily solutions, is the most recent GRACE-only gravity field model computed in Graz (Mayer-Gürr et al., 2018).*

5. Section 3.2, how do you evaluate/consider the reliability of the GESD test, please give a more detailed demonstration.

**Response**: We thank reviewer's suggestions. GESD test is a commonly used univariate anomaly detection algorithm. It has been widely used in the field of hydrological anomaly detection (Saghafian et al., 2014; Clark and Zipper, 2016).GESD test is mainly used in this study to extract possible flood days corresponding to the high-frequency signals. In this study, the method to be selected for extracting flood information from the high-frequency signals should ensure that it was affected by the random error in the high-frequency signal as little as possible and that the flood signals were extracted as much as possible. The GESD test method overcomes the primary limitation of the Grubbs test and the Tietjen-Moore test that the suspected number of outliers, k, must be specified exactly. The GESD test only requires that an upper bound for the suspected number of outliers be specified (Rosner, 1983). These advantages of the GESD test method were highly compatible with our needs.

Moreover, in the practical operation for our dataset, the method performed well. The method effectively extracted the important anomalous peaks (i.e., the pre-extracted

flood days), and the extracted flood days for each spatial grid were stable and did not increase with the increase of the preset up bound. As shown in Fig. 2, the histogram distribution of the pre-extracted global flood days for each spatial grids was concentrated around 200 days, not more than 500 days. It was not more than 200 flood events after converted from flood days to flood events. The order of magnitude was consistent with the number of historical flood events. Finally, the pre-extracted flood based on GESD test was an important part of the final extracted flood days, and the good validation of the final product also confirmed that GESD test method played a reliable and important role in the intermediate processing.

We have added a more detailed demonstration in Lines 22-30, Pages 1-2 in supplementary file.

*We also considered the reliability of the GESD test. The method effectively extracted the important anomalous peaks (i.e., the pre-extracted flood days) and the extracted flood days for each spatial grid were stable and did not increase with the increase of the preset up bound in our study. As shown in Figure S4, the histogram distribution of the pre-extracted global flood days for each spatial grids was concentrated around 200 days, not more than 500 days. It was not more than 200 flood events after converted from flood days to flood events. The order of magnitude was consistent with the number of historical flood events (shown in results section). The advantages of the GESD test method and good performance on the flood days extraction laid a solid method foundation for subsequent analysis in this study.*

[Figure]

Figure S4 The histogram of pre-selected flood days based on high-frequency signals of TWS using GESD test method.

6. When I check the supplementary materials, I find that there are many differences among the spatial patterns of DFO-based, MODIS-based, and GRACE-based floods. More discussion should be drawn to improve this MS, the work in Gouweleeuw (2018) also showed that ITSG might not catch the flooded area well. Besides, how do you consider the different representation between terrestrial water storage (TWS) and runoff for one flood event?

**Response**: We thank reviewer's suggestions. The DFO database mainly records large-scale flood events from news and government announcements and misses some flood events. Although DFO records the start and end times and the approximate spatial locations of flood events, the time duration is sometimes long (more than one or two months), and the spatial locations are only roughly delineated according to news reports, which contained large uncertainties in spatial distribution and duration. Moreover, although flood detection methods based on MODIS data could finely delineated the spatial inundation ranges, it was mainly aimed at specific flood events in small areas. Only a limited amount of flooding can be identified by remote sensing images due to the influence of bad weather. MODIS resolution is 1 km, which is a higher resolution product compared to GRACE TWS (1°, ~100km). The GRACE-based flood days we provided can only indicate the presence of flood events under the ~100km grid coverage, and specific detailed locations require further identification by high-resolution satellite remote sensing image. Therefore, we can see differences among the spatial patterns of the three datasets. However, as we show in Figure 8-9 and supplementary figures, there is still an overwhelming majority of results indicating a consistent spatial pattern of flood inundation. Our products can provide a sufficient data foundation for large-scale research on the spatiotemporal distributions and attributions of global floods. We have added detailed discussion in Discussion section in Lines 383-389, Page 16.

*The flood extents recorded by DFO are rough and time durations are sometimes long (much more than 1 month), which contained large uncertainties in spatial distribution and duration. Although MODIS resolution (1 km) is higher than that of GRACE TWS (1 °, ~100km), only a limited amount of flooding can be identified by remote sensing images due to the influence of bad weather. The GRACE-based flood days we provided was only able to indicate the presence of flood events under the ~100km grid coverage, and specific detailed flood extents require further identification by high-resolution satellite remote sensing image. Therefore, we can see some differences among the spatial patterns of flood recorded by DFO, MODIS and GRACE.*

As for the different representation between terrestrial water storage (TWS) and runoff for one flood event, we extracted the flood days based on GRACE. To facilitate the comparison between GRACE-based floods and discharge-based floods, we first extracted flood events from discharge data and merged them according to whether there was an intersection in the time series in the same river basin. Then we compared

whether each flood event in the river basin contained the GRACE-based flood days. We have added it in Lines 421-430, Page 17.

*We focused on extreme precipitation-induced flood events and similarly constrained the results derived from the discharge data with extreme precipitation data. To ensure accuracy, we first selected the floods extracted from discharge stations with GFS greater than 0.5 for comparison. Due to discharge reflects the amount of water integrated over its entire contributing basin and contributing time (Yang et al., 2019), we combined the flood events obtained from each discharge station in time series to describe the flood events in the 261 watershed (HydroSHEDS Basins Level 4 (Lehner and Grill, 2013; Lehner et al., 2008)). Flood events in the same watershed were merged according to whether there was an intersection in the time series. The accuracy index used for comparison in this study is the probability of detection (POD) (Yang et al., 2021), i.e., whether each flood event in the river basin contained the GRACE-based flood days. Although flood events derived from discharge cannot guarantee that the surrounding land will experience flooding, it provides us a reference to support the reliability of the time series verification process.*

Minor Comments:

1. Figure 3, the tick values of the x-axis should be denser.

**Response**: We thank reviewer's suggestions. We have made the x-axis denser.

2. P2L69ï¼ what is the "other places"?

**Response**: We thank reviewer's suggestions. We have changed the "other" to a more critical description in Lines 67-68, Page 2.

*The numbers of flood events recorded in China, Russia and Canada are obviously lacking.*

3. "Diksha Gupta et al (Gupta and Dhanya, 2020)"… in all the MS, these citation formats are strange, maybe wrong.

**Response**: We thank reviewer's suggestions. We have modified it to the correct format.

4. P3L109, what is the difference between the two citations of "Dill, 2008; Dill et al.,

**Response**: We thank reviewer's suggestions. One was misquoted here and has been removed.

5. P10L296, "Tellman1" to "Tellman"

**Response**: We thank reviewer's suggestions. We have re-written it.

6. The two references share the same data name, I think there is no need to put them on two websites. In addition, there is a mistake in the name of the authors.

"Zhang, J., Liu, K., and Wang, M.: Flood detection using GRACE Terrestrial Water Storage and Extreme Precipitation (1.0.0). Zenodo. https://doi.org/10.5281/zenodo.6831105, 2022a.

Zhang, j., Liu, K., and Wang, M.: Flood Detection Using GRACE Terrestrial Water Storage and Extreme Precipitation (1.0.0) [Data set]. Zenodo. https://doi.org/10.5281/zenodo.6831384, 2022b."

**Response**: We thank reviewer's suggestions. Since each uploaded file generates a unique DOI, we couldn't put them on one website. We have modified the mistake in the name of the authors.

7. I do not think the reference format is right. "Mayer-Gürr, T., Behzadpour, S., Kvas, A., Ellmer, M., Klinger, B., Strasser, S., and Zehentner, N.: ITSG-Grace2018: monthly, daily and static gravity field solutions from GRACE, 2018."

**Response**: We thank reviewer's suggestions. We have modified this reference format.

8. The references should be checked carefully.

**Response**: We thank reviewer's suggestions. We have checked all citation formats of this paper.

**Response to reviewer #3**

1.  Why is the MODIS-based flood dataset used for comparison only 807 flood events and much smaller than the 2380 flood events recorded in the DFO database?

**Response**: We thank reviewer's suggestions. The MODIS-based flood dataset was extracted by B. Tellman et al. based on the DFO-recorded flood. Due to the influence of weather conditions, the author only extracted 913 large-scale MODIS-based flood events from 2000 to 2018 (Tellman et al., 2021). Therefore, this study used 807 flood events from April 1st, 2002 to August 31st, 2016.

2.  Please provide details on how to specifically compare one discharge-based flood and one TWS-based flood in this paper.

**Response**: We thank reviewer's suggestions. The final products in this study were flood days. To facilitate the comparison between GRACE-based floods and discharge-based floods, we first extracted flood events from discharge data and merged them according to whether there was an intersection in the time series in the same river basin. Then we compared whether each flood event in the river basin contained the GRACE-based flood days. We have added it in Lines 421-429, Page 17.

*We focused on extreme precipitation-induced flood events and similarly constrained the results derived from the discharge data with extreme precipitation data. To ensure accuracy, we first selected the floods extracted from discharge stations with GFS greater than 0.5 for comparison. Due to discharge reflects the amount of water integrated over its entire contributing basin and contributing time (Yang et al., 2019), we combined the flood events obtained from each discharge station in time series to describe the flood events in the 261 watershed (HydroSHEDS Basins Level 4 (Lehner and Grill, 2013; Lehner et al., 2008)). Flood events in the same watershed were merged according to whether there was an intersection in the time series. The accuracy index used for comparison in this study is the probability of detection (POD) (Yang et al., 2021), i.e., whether each flood event in the river basin contained the GRACE-based flood days.*

3.  Why is the threshold value set to 0.1 in the GESD method? Please add a detailed explanation.

**Response**: We thank reviewer's suggestions. The selection of this preset threshold (0.1) could not affect the actual number of outliers detected by the algorithm. Considering that GESD can detect abnormal days adaptively, it should be ensured that the number

of outliers actually detected is less than or equal to the preset threshold. As shown in Figure 1, 0.1 is selected here to ensure that the number of outliers in each grid in the world does not exceed the preset number (i.e. 527) and the number of possible flood days will not be missed. We have added a detailed explanation in Lines 22-30, Pages 1-2 in supplementary file.

*We also considered the reliability of the GESD test. The method effectively extracted the important anomalous peaks (i.e., the pre-extracted flood days) and the extracted flood days for each spatial grid were stable and did not increase with the increase of the preset up bound in our study. As shown in Figure S2, the histogram distribution of the pre-extracted global flood days for each spatial grids was concentrated around 200 days, not more than 500 days. It was not more than 200 flood events after converted from flood days to flood events. The order of magnitude was consistent with the number of historical flood events (shown in results section). The advantages of the GESD test method and good performance on the flood days extraction laid a solid method foundation for subsequent analysis in this study.*

[Figure]

Figure S2 The histogram of pre-selected flood days based on high-frequency signals of TWS using GESD test method.

**4.   Why is only the area within 60 degrees north and south latitude shown?**

**Response**: We thank reviewer's suggestions. The selection of the area was mainly determined by the range of GPM data. The GPM data mainly covers the range of 60°S—60°N. Therefore, we selected the range of 60 degrees north-south latitude of the GRACE data to be consistent with GPM.

5.  Please indicate the number of available or used discharge measurements.

**Response**: We thank reviewer's suggestions. We used in total 3408 discharge stations from GRDC around the world. We have added this in Lines 149-152, Page 4.

*The unit of mean daily discharge is $m^3/s$ and the stations with more than 50% missing days in research time period (Apr. 1st, 2002—Aug. 31st, 2016) were excluded to ensure the accuracy. Finally, we obtained 3408 stations from Apr. 1st, 2002—Aug. 31st, 2016 as the validation dataset to verify the GRACE-derived flood days.*

6.  Why did you choose maximum values of the precipitation data under the GRACE grid coverage in the process of harmonizing both resolution?

**Response**: We thank reviewer's suggestions. We mainly focus on flood events caused by extreme precipitation, so effectively extracting extreme precipitation signals is the basis for ensuring the integrity of flood event extraction. Considering the regional differences of precipitation at GRACE resolution level, the maximum precipitation under the GRACE grid can retain the signal of extreme precipitation to the greatest extent. We have also tried to take the mean value of the precipitation corresponding to the GRACE grid coverage, but this leads to a missing of many flood events. We have added this in the Discussion section in Lines 499-502, Page 21.

*Second, considering the regional differences of precipitation at GRACE resolution level, the maximum precipitation under the GRACE grid can retain the signal of extreme precipitation to the greatest extent. We have also tried to take the mean value of the precipitation covered by GRACE grid, but this led to many missing flood events.*

7.  Please add units to the precipitation and GRACE TWS data separately

**Response**: We thank reviewer's suggestions. We have added the units to the precipitation and GRACE TWS data description in Lines 106-107, Page 3 and Lines 116-117, Page 3.

*The time period spans from Apr. 1st, 2002, to Aug. 31st, 2016, the resolution is 1° and the unit is "m".*

*The resolution of these data is 0.1°, with the unit of "mm", mainly covering the range of 60°S—60°N, and both north-south latitudes of 60—90° have partial coverage.*

**Response to reviewer #4**

The paper titled "Flood Detection Using GRACE Terrestrial Water Storage and Extreme Precipitation" by lead author Jianxin Zhang and co-authors explored capacity of the combination of satellite products, the GRACE total water storage and GPM precipitation, for detection of flood events between 60°S and 60°N. The authors have done an considerable efforts for validation of obtained results (flood location and duration) against existing flood records collected by Dartmouth Flood Observatory, MODIS-derived flood product, social media records as well as flood events derived by the authors from the GRDC river discharge records. The idea is interesting and the theoretical workflow seems to be consistent with the objectives.

My major concern with the paper is the lack of clarity and consistency in the description of the datasets used and in validation of new flood product. The results of validation of the GRACE derived product are mostly qualitative and does not allow for evaluation of the overall quality. Moreover, the structure of sentences, unnecessary repetitions and the vocabulary used make the reading and understanding difficult. I would recommend to re-write the manuscript with the help of a native English speaker who is also an expert in the Remote Sensing. I am also not convinced of the value of the flood event product of such a low 1°x1° spatial resolution.

Overall, I think the paper needs a major revision.

**General comments.**

1. Please, re-write the description of the datasets and products used. Some sentences give an impression that the authors did an additional transformation of GRACE and MODIS products, but I could not understand what was done by teams elaborated/provided the products and what was added by the authors of current manuscript. What was the source of the GRACE daily product? Please, provide the link. Does the link for the DFO flood dataset exist? The GRDC discharge data description is inadequate. The social media flood events database description is given in the Result section and its description lacks the details (number of recorded floods, spatial and temporal coverage etc). I also advice to pay an attention on the titles of sections (for example, sec. 2.1, 2.3). The section 3.3 needs to be carefully re-written. In the section 3.5 an important information about FloodR is missing (approach used in the package, realisation, validation accuracy).

**Response:** We thank reviewer's suggestions. We have re-written the description of the datasets and products used. Our specific responses to each question are as follows:

1) What was the source of the GRACE daily product? Please, provide the link.

We have provided the source of the GRACE daily product in Lines 109-110, Page 3 in the revised manuscript.

*This processed data can be obtained from the website: https://www.tugraz.at/institute/ifg/downloads/gravity-field-models/itsg-grace2018/.*

2) Does the link for the DFO flood dataset exist?

Yes, the link for the DFO flood dataset exists and we have provided it in Lines 131, Page 4 in the revised manuscript.

*This product was primarily used to validate the flood data extracted in this study and can be obtained from https://floodobservatory.colorado.edu/Archives/index.html.*

3) The GRDC discharge data description is inadequate.

We have elaborated the description of GRDC discharge data in Lines 145-152, page 4 in the revised manuscript.

*The Global Runoff Data Centre is an international data centre operating under the auspices of the World Meteorological Organization. It was established in 1988 to support research on global climate change and integrated water resource management. We downloaded the global mean daily discharge data from the https://www.bafg.de/GRDC/EN/Home/homepage_node.html, which additionally contained other attributes like country, longitude, latitude and river name associated with each flood event. The unit of mean daily discharge is $m^3$/s and the stations with more than 50% missing days in research time period (Apr. 1st, 2002—Aug. 31st, 2016) were excluded to ensure the accuracy. Finally, we obtained 3408 stations from Apr. 1st, 2002—Aug. 31st, 2016 as the validation dataset to verify the GRACE-derived flood days.*

4) The social media flood events database description is given in the Result section and its description lacks the details (number of recorded floods, spatial and temporal coverage etc).

As we mentioned in the main text, DFO is a Global Active Archive of Large Flood Events derived news, governmental, instrumental, and remote sensing source, and some flood events were inevitably missing. In this study, we randomly selected some flood events from news to test whether additional flood events could be identified, as shown in Figure 15. But we were unable to obtain all additional flood events not recorded by DFO. We have re-written it to make it clearer in Lines 407-415, Pages 16-17 in the revised manuscript.

*We further selected some flood events through news reports and social medias (like Twitter, Weibo) to verify if there were some flood events that couldn't be identified by the DFO but can be identified by GRACE-based flood days. Figure 15 presented 9 flood events not recorded by the DFO, including floods that occurred in different time periods and areas such as the eastern United States, northern and southern South America, Mozambique in Africa, France, India, China, Malaysia, Indonesia and Australia. The red boxes indicated the approximate location of the reported flood events. The blue areas were the GRACE-based flood days corresponding time duration and approximate location of flood events recorded by social media or news reports. We found that GRACE-based flood days could well identify these missing flood events, which also proved the effectiveness of using GRACE to identify large-scale flood events. Our data can be used as a good supplement to DFO data.*

5) I also advice to pay an attention on the titles of sections (for example, sec. 2.1, 2.3)

We have renamed the titles of sections 2.1 and 2.3 as follows:

*2.1 Daily GRACE TWS*

*2.3 Flood events from Dartmouth Flood Observatory*

6) The section 3.3 needs to be carefully re-written.

We have re-written the section 3.3 as follows:

*We used the probable flood days extracted by flood potential index (FPI) as a complement to the inability to detect flood events with high-frequency signals. The FPI mainly considered rainfall-induced floods and has been widely used to evaluate flood events (Gupta and Dhanya, 2020; Molodtsova et al., 2016; Reager et al., 2014). Its basic assumption is that the regional water storage capacity can be approximated*

*by the maximum value of historical TWS time series. The water storage capacity at*

*the current time can be calculated by subtracting the TWS at the previous time from the maximum value of TWS time-series. The proposal of this method was based on monthly data, but this does not affect the application of GRACE daily data. The detailed descriptions were as follows.*

*The water storage capacity of the current day can be expressed as the temporal difference between the maximum time-series value and the previous-day value, and the formula is expressed as follows:*

$$TWS_{DEF}(t) = TWS_{MAX} - TWS(t-1) \qquad (4)$$

*where $TWS_{DEF}(t)$ represents the maximum allowable relative water storage change on the current day, $TWS_{MAX}$ represents the maximum value over the entire time series, and $TWS(t-1)$ represents the TWS value of the current day relative*

*to the previous day. A low storage deficit and high precipitation mean a high probability of flooding, i.e., the occurrence of floods should be based on the mismatch between the extreme precipitation level and the increase in water storage:*

$$F(t) = P_{day}(t) - TWS_{DEF}(t) \tag{5}$$

*where $P_{day}(t)$ represents the daily precipitation and $F(t)$ represents whether the current precipitation matches the water storage capacity. When $F(t) > 0$, flooding may occur. This study uses the flood potential index to supplement possible flood days in case the daily GRACE TWS data have lost useful high-frequency signals due to the interpolation process. We have taken an example in Figure S3 to present flood potential index was able to supplement some flood events not identified by GRACE high-frequency signals.*

**Supplementary materials:**

*Figure S3 showed an example that flood potential index was able to supplement some flood events not identified by GRACE high-frequency signals. Red rectangular boxes indicated floods detected by FPI but not detected by high-frequency signals.*

[Figure]

**Figure S3 Flood potential index supplemented floods unrecognized by GRACE TWS high-frequency signals (red rectangular boxes).**

7) In the section 3.5 an important information about FloodR is missing (approach used in the package, realisation, validation accuracy).

We have re-written the section 3.5 as follows:

*To verify the reliability of the extracted results, this paper used the global discharge data products released by GRDC and the statistics-based automated flood event separation (FloodR) method to extract flood events. FloodR is a statistical-based flood event separation method proposed by Fischer et al. (2021). It can automatically separate flood events using a univariate daily discharge time series, and it integrates expert knowledge tools to manually and quickly validate and correct the separation results. Considering that the fluctuation of daily discharge data is smoother than that of hourly discharge data, FloodR used the moving-window variance to overcome the lower dynamic characteristics of daily discharge. It's basic rules including three points: 1) a flood event is an event that temporarily exceeds the normal discharge, and the start and end of each flood event can be defined; 2) a flood event can be*

*characterized by significantly increased dynamics of discharge; 3) the sum of the increasing discharges is similar to the sum of the recession of the flood event (Fischer et al., 2021). FloodR can also automatically handle missing data, and perform flood separation in segments according to the missing data and finally merge them. In this paper, the function "eventsep" in FloodR package was used and the parameters was default according to the results of author's practice while the parameter "NA_mode" is based on whether there are missing values in the discharge time series. The results extracted by FloodR include information like the start and end times of each flood, the flood peak date and the flood baseflow, etc., thus providing an important data foundation for verifying the time-series comparison ability of this study.*

*We use goodness of flood separation (GFS) to evaluate the performance of FloodR method. This indicator explicitly minimizes the number of small runoff events and maximizes the number of flood events with high discharge. This indicator can be used to address the situation of lacking a consistent and true data foundation for evaluate the goodness of flood separation (Fischer et al., 2021).*

$$\text{GFS} = \left(\frac{Q_{Q>TH_{upper};Flood}}{Q_{Q>TH_{upper}}}\right) - \max(\frac{Q_{Q>TH_{lower};Flood}}{Q_{Q>TH_{lower}}} - \text{Tol}_{lower}, 0) \tag{6}$$

*Where $Q_{Q>TH_{upper};Flood}$ is the number of flood days with discharge above the threshold of $TH_{upper}$; $Q_{Q>TH_{upper}}$ is the number of days above the threshold of $TH_{upper}$; $Q_{Q>TH_{lower};Flood}$ is the number of flood days with discharge below the threshold of $TH_{lower}$; $Q_{Q>TH_{lower}}$ is the number of days below the threshold of $TH_{lower}$. The upper threshold $TH_{upper}$, lower threshold $TH_{lower}$ and tolerance threshold $Tol_{lower}$ are set 95% quantile, 50% quantile and 1% according to author's suggestion (Fischer et al., 2021).*

2. Moreover, the authors often confuse the concepts of method, approach and product. For the citations, the use of surname, name and affiliation followed by (Name et.al., 20XX) is not a common practice.  Please, consult the ESSD journal citation model.

**Response**:We thank reviewer's suggestions. We have checked the difference between approach and method. Approach is the way you are going to approach the project and Method is the way in which you are going to complete the project. In the revised manuscript, we have made a correction throughout the paper. The citation in this paper were also corrected according the requirement of ESSD.

3.  Several software, tools or codes were used for processing of data. The description of these tools is inappropriate. A short physical/ mathematical description of mentioned parameters (such as "t.window" and "s.window", "direction") is required.

**Response:** We thank reviewer's suggestions. We have added a physical/ mathematical description of mentioned parameters in Lines 170-176, page 5; Lines 200-207, Page 6.

*In this study, the STL function in the R language "stats" package was used to process all grid time series corresponding to the GRACE TWS period (Apr. 1st, 2002—Aug. 31st, 2016). The two main parameters, "t.window" and "s.window", should be specified when using STL. "t.window" is the number of consecutive observations when estimating the trend-cycle and it was set to 31-day window to cover the month and separates daily data according to Gouweleeuw et al. (2018) and Xiong et al. (2022). "s.window" is the number of consecutive years when estimating each value in the seasonal component and it was set to 360 which was determined using a Fourier transform to convert to the frequency domain to obtain the frequency corresponding to the maximum amplitude.*

*This package not only includes the GESD algorithm but can also specify the direction of detected outliers. The parameter "direction" indicates whether to extract peaks or valleys and "pos" means the extraction of peaks, and "neg" means the extraction of valleys. As we considered extreme weather events caused by heavy precipitation in this study, important information was contained in the peak. The main parameter "direction" was set to the "pos", and the maximum possible number of abnormal days "max_anoms" was set to 0.1 to cover the maximum number of abnormal days among the global time series comprising every grid.*

4.  Regarding the validation of the obtained results, the authors compare the number of detected GRACE-precipitation derived flood events with the DFO database and provide the figures and some light statistics. They also refer to the comparison of their flood event retrievals with the MODIS-derived dataset. A large archive of figures is downloadable as supplementary materials, however the statistical evaluation of comparison of GRACE and MODIS flood events is missing in the text. I would appreciate a summary table with statistics for comparison with all 4 reference products (DFO dataset, MODIS flood events, discharge-derived flood events and social media flood records). Important validation information is scattered throughout the text, but the quality of the language does not allow to understand well the authors' logic in many subsections.

**Response:** We thank reviewer's suggestions. The statistical evaluations were listed (except social media flood records) in the summary table in Lines 452-456, page 19.

*We have listed the flood detection performance compared with DFO, MODIS and discharge respectively in the Table 1. GRACE was able to detect 81% of flood events*

recorded by DFO and 87% of flood events recorded by MODIS. If we summed all flood events from 261 river basins, GRACE-based flood days could identify 53% flood events derived from discharges. The percentage of river basins with POD values greater than or equal to 0.5 reached 62%.

**Table 1 Flood detection performance compared with DFO, MODIS and discharge data**

|  | DFO (number of flood events) | MODIS (number of flood events) | Discharge (number of flood events) | Discharge (number of river basins) |
|---|---|---|---|---|
| Total | 2380 | 807 | 10472 | 261 |
| Detection | 1917 | 703 | 5597 | 156 (POD>=0.5) |
| Percent | 81% | 87% | 53% | 62% |

As we mentioned in the main text, DFO is a Global Active Archive of Large Flood Events derived news, governmental, instrumental, and remote sensing source, and some flood events were inevitably missing. In this study, we randomly selected some flood events from news to test whether additional flood events could be identified, as shown in Figure 11. But we were unable to obtain all additional flood events not recorded by DFO. We have re-written it to make it clearer in Lines 407-415, Pages 16-17 in the revised manuscript.

*We further selected some flood events through news to verify if there are some flood events that cannot be identified by the DFO but can be identified by GRACE-based flood days. Figure 11 presented 9 flood events not recorded by the DFO, including floods that occurred in different time periods and areas such as the eastern United States, northern and southern South America, Mozambique in Africa, France, India, China, Malaysia, Indonesia and Australia. The red box indicated the approximate location of the reported flood events. The blue areas were the GRACE-based flood days in this study. We could find that GRACE-based flood days have well identified these missing flood events, which also proved the effectiveness of using GRACE to identify large-scale flood events. Our data can be used as a good supplement to DFO data.*

**Specific comments.**

5.  Line 55. Repeated information.

**Response:** We thank reviewer's suggestions. We have re-written it in Lines 55-57, Page 2.

*However, there are some limitations in current databases. NatCatSERVICE, EM-Dat and Sigma provide data only at the country level. The NatCatSERVICE database covers*

*most large flood events around the world but only a few small flood events in developing countries due to restricted connectivity (De Bruijn et al., 2019).*

6. Lines 63 and 66. Replace the word "range".

**Response:** We thank reviewer's suggestions. We have replaced "temporal range" with "time duration" and replaced the second "range" with "extent" in Lines 63 and 66, Page 2.

*Although DFO records the start and end times and the approximate spatial locations of flood events, **the time duration is sometimes long** (more than one or two months), and the spatial locations are only roughly delineated according to news reports. B. Tellman et al. (2021) extracted flood extents and analysed the population exposure of 913 large-scale flood events from 2000 to 2018 based on MODIS daily data with a resolution of 250 metres, thus finely delineating the **spatial inundation extent.***

7. Line 68. Repeated information

**Response:** We thank reviewer's suggestions. We have deleted sentence **"Moreover, only some large-scale flood events were recorded by the DFO."**

8. Line 72. "Much useful image information" - be more specific. What kind of information is missing?

**Response:** We thank reviewer's suggestions. The spectral information of optical remote sensing image is influenced by clouds, which affects the quantitative inversion of flood extent based on remote sensing. SAR image lacks revisits of the same location for flood change detection. We have added this explanation in revised manuscript in Lines 70-73, Page 2.

*The spectral information of optical remote sensing image is influenced by clouds, which affects the quantitative inversion of flood extent based on remote sensing. SAR image lacks revisits of the same location for flood change detection. These shortcomings affected the flood extraction accuracy.*

9. Line 78-79. What the "multiyear flood observation data" is?

**Response:** We thank reviewer's suggestions. The " *multiyear flood observation data* " means "*multiyear flood observation data from 2003 to 2012 by the US Geological Survey and DFO*". We have re-written the description in Lines 77-79, Page 2.

*Molodtsova et al. (2016) found an agreement between the flood potential index derived from GRACE and recorded floods by using multiyear flood observation data from 2003 to 2012 by the US Geological Survey and DFO.*

10. Line 81 and 85. Specify the method in the "this method"

**Response:** We thank reviewer's suggestions. We have specified "this method" with "flood potential index" in Line 82, Page 2.

*They suggested that **flood potential index** can be useful for flood monitoring when discharge data are rarely available.*

11. Lines 112-115. The sentence is too long.

**Response:** We thank reviewer's suggestions. We have separated the long sentence into short ones to make it clearer in Lines 114-117, Page 3.

*It is the next-generation and high-quality global rain and snow satellite observation network after the TRMM. GPM provides important data foundation for scientific researchers to understand the Earth's water resources and energy cycles and improve their ability to predict extreme events (Huffman et al., 2015).*

12. Lines 120. What does it mean " under the GRACE grid coverage"?

**Response:** We thank reviewer's suggestions. It means we selected the maximum value of precipitation grid covered by each GRACE grid to maintain the extreme precipitation information. We have made it clearer in Lines 121-123, Page 4

*To remain consistent with the GRACE resolution and maintain extreme precipitation signals, we take the maximum values of the precipitation covered by each 1° GRACE grid to further calculate the flood potential index and the number of extreme precipitation days.*

13. Lines 134-137. The sentence is too long. What is the "Otsu" ?

**Response:** We thank reviewer's suggestions. We have separated the long sentence into short ones to make it clearer. The "Otsu" means Otsu-optimized thresholds method. We have re-written it and added the reference in Lines 136, Page 4.

*This product was produced based on atmospherically corrected Terra (MOD09GA/GQ) and Aqua (MYD09GA/GQ) MODIS images. Then the authors used threshold analysis methods (including standard and Otsu-optimized thresholds methods) and slope constraints (slopes greater than 5° were masked out) to extract inundations at a 250-m spatial resolution according to the flood events recorded by the DFO (Tellman et al., 2021).*

14. Line 137 and in all other places. Please, replace the phrase "extraction results" with more specific terms.

**Response:** We thank reviewer's suggestions. We have replaced the phrase "extraction results" with " MODIS-based floods" in Line 138, Page 4.

*The **MODIS-based floods** were compared and verified for coincidence with the 30-m-resolution inundation data derived from Landsat 5, 7 and 8 imageries, and flood map quality control analysis was also performed.*

15. Line 154. The sentence is too banal.

**Response:** We thank reviewer's suggestions. We have re-written the banal sentence in Lines 154-155, Page 4.

*Figure 1 shows the technical route of this study. It mainly consists of data preparation, flood days extraction, and result verification.*

16. Line 158. Find another term for "preliminary possible flood dates"

**Response:** We thank reviewer's suggestions. We have modified it as "pre-selected possible flood days" in Line 157 Page 4.

*The flood extraction step is mainly based on high-frequency signals of TWS and the flood potential index to obtain the **pre-selected possible flood days**; then, extreme precipitation constraints are used to obtain the final flood days.*

17. Line 159. Not clear the comparison of what with what was done during the validation described here. If I understood well the section 2.3, the flood extent is not provided in DFO dataset.

**Response:** We thank reviewer's suggestions. DFO contains the approximate flood extent. To avoid confusion, we have elaborated the description of DFO data in the section 2.3 in Lines 125-127, Page 4; Lines 158-160, Pages 4-5.

*The flood validation includes comparisons with the DFO-recorded flood extent, MODIS-derived flood inundation, GRDC discharge-derived flood events and significant flood events recorded on social media.*

*The DFO dataset records large flood events from various news reports and government websites. It contains the start and end times of each flood, the country where it occurred, the approximate flood extent, the cause of the flood and the degree of damage.*

18. Line 168. The phrase "to obtain the season" is the scientific slang.

**Response:** We thank reviewer's suggestions. We have re-written the sentence in Lines 165-167, Pages 5.

*Seasonal and trend decomposition using loess (STL) (Robert et al., 1990) is a filtering process as well as a general and robust time series decomposition and forecasting method used to decompose time series variables into seasonal, trend and remainder components for further forecasting.*

19. Line 169. Please, explain what the "high-frequency seasonal data" is?

**Response:** We thank reviewer's suggestions. The term 'seasonal' should be 'signal' and we have corrected it in Lines 168, Page 5.

*This process can handle data with any type of seasonality as well as **high-frequency signal data.***

20. Lines 198-201. Please, provide the physical meaning for term "direction". The phrase ..."direction" was set to the position..." is not clear.

**Response:** We thank reviewer's suggestions. We have provided physical meaning for term "direction" in Lines 201-205, Pages 6 to make it clearer.

*The parameter "direction" indicated whether to extract peaks or valleys and "pos" means the extraction of peaks, and "neg" means the extraction of valleys. As we considered extreme weather events caused by heavy precipitation in this study, important information was contained in the peak. The main parameter "direction" was set to the "pos", and the maximum possible number of abnormal days "max_anoms" was set to 0.1 to cover the maximum number of abnormal days among the global time series comprising every grid.*

21. Lines 203-205. Repeated information.

**Response:** We thank reviewer's suggestions. We have removed the repeated information and re-written it in Lines 209-211, Page 7.

*We used the probable flood days extracted by flood potential index (FPI) as a complement to the inability to detect flood events with high-frequency signals. The FPI mainly considered rainfall-induced floods and has been widely used to evaluate flood events (Gupta and Dhanya, 2020; Molodtsova et al., 2016; Reager et al., 2014).*

22. Line 206. Rephrase the first part of the sentence.

**Response:** We thank reviewer's suggestions. We have rephrased the first part of sentence in Lines 209-215, Page 6.

*We used the probable flood days extracted by flood potential index (FPI) as a complement to the inability to detect flood events with high-frequency signals. The FPI mainly considered rainfall-induced floods and has been widely used to evaluate flood events (Gupta and Dhanya, 2020; Molodtsova et al., 2016; Reager et al., 2014). Its basic assumption is that the regional water storage capacity can be approximated by*

*the maximum value of historical TWS time series. The water storage capacity at the*

*current time can be calculated by subtracting the TWS at the previous time from the maximum value of TWS time-series. The proposal of this method was based on monthly data, but this does not affect the application of GRACE daily data. The detailed descriptions were as follows.*

23. Line 199. The issue with the missing of the high-frequency signal in the daily GRACE product needs more details in the section 2.1. and some discussion in the Discussion section.

**Response:** We thank reviewer's suggestions. We have made a supplement in revised manuscript in supplementary file in Lines 9-20, Page 1, some descriptions in Lines 205-207, Page 6 and some discussion in the Discussion section in Lines 503-505, Page 21.

*In Figure S1, We took a GRACE grid in China as an example to show that high frequency signals may loss some useful information and this was the reason why we considered flood potential index as a supplement to missed flood events by high frequency signals. We obtained the high-frequency signals and analyzed the DFO flood event covering this grid. The unrecognizable flood of GRACE high-frequency signals was due to the fact that AR interpolation was used in GRACE TWS and it introduces white noise. This white noise will be filtered out in the subsequent GESD test method, resulting in the loss of useful signals (red rectangular boxes). The other reason of unrecognition was that GRACE TWS itself was not able to identify the flood event (blue rectangular boxes).*

[Figure]

Figure 5 The high-frequency signal of GRACE-id-4803 grid and flood detection. Red rectangular boxes indicate unrecognized floods due to GRACE high-frequency signals, and blue rectangular boxes indicate unrecognized floods due to GRACE missing months.

*We have taken an example in Figure S1 to show the process of extracting possible flood days from high-frequency signals of GRACE TWS using GESD test method as well as the reason for missing some flood events. We have also considered the reliability of the GESD test method in Figure S2.*

*Third, the high frequency signals of GRACE TWS may loss some flood events as demonstrated before. Although FPI can supplement some flood events that were not identified by high-frequency signals, it can not guarantee that all flood events lost due to high-frequency signals could be supplemented.*

24. Lines 222-223. Please, do not repeat the banal phrases in the beginning of each section, provide more specific for each section information instead.

**Response:** We thank reviewer's suggestions. We have re-written the beginning of each section in Lines 230-232, Page 7.

*The flood extraction mainly went through pre-selection stage and final selection stage. We first used GRACE HPF data combined with GESD method and flood potential index to preselect the possible flood days pixel by pixel. Next, we further used the number of extreme precipitation days to constrain and obtain the final flood days.*

25. Line 224. The floods are not "affected" by precipitation, but can be "caused".

**Response:** We thank reviewer's suggestions. We have corrected the word with " caused" in Lines 233, Page 7.

*Considering that floods are **caused** not only by single-day precipitation but also by cumulative precipitation, we calculated the extreme precipitation days based on the one-day precipitation, 3-day cumulative precipitation and 5-day cumulative precipitation.*

26. Line 234. Please, replace the word "separation"

**Response:** We thank reviewer's suggestions. We have replaced the word "separation" with "extraction".

27. Line 240. What is the flood "baseline"?

**Response:** We thank reviewer's suggestions. The word "baseline" means "baseflow", To avoid confusion, we have changed "baseline" as "baseflow".

28. Figure 5 and in the text. Replace the "polygon feature" with the name of parameter/phenomena represented by these polygons.

**Response:** We thank reviewer's suggestions. We have replaced the "polygon feature" with "DFO-recorded flood events" in Line 393, Page 18

*Figure 8 Spatial distribution of DFO-recorded flood events*

29. Line 279. Some part of the sentence is missing.

**Response:** We thank reviewer's suggestions. We have re-written the sentence in Lines 362-363, Page 15.

*In this study, the temporal length of the DFO database was compared with GRACE-based flood throughout the Apr. 1st, 2002—Aug. 31st, 2016 period.*

30. Line 287-288 and the Figure 6. are not clear.

**Response:** We thank reviewer's suggestions. We have re-written the sentence and made the Figure 6 clearer in Lines 370-371, Page 15.

*Figure 9 showed the distribution of the number of flood events recorded by DFO on spatial 1-degree grids (the same as GRACE spatial resolution).*

[Figure]

Figure 9 The frequency of flood events recorded by DFO distributed on 1-degree grids.

31. Line 347. " ...the spatial average of the discharge data under the coverage of the HydroSHEDS Basins Level 4 data". The spatially averaged discharge is something new in hydrology. I would ask the authors to provide a reason to invent this parameter, explain its physical meaning and give a solid base for its application in the context of the study. For me the "spatially averaged discharge" is something meaningless.

**Response:** We thank reviewer's suggestions. Following the reviewer's suggestion, instead of using "spatially averaged discharge", we calculated the flood events of each discharge station separately for comparison with GRACE-based flood days. We have re-written it in Lines 419-456, Pages 17-19.

*We also compared our results with discharge data to assess our detection ability. We used the FloodR method of Fischer et al. (2021) to extract flood events from 3408 GRDC discharge data to serve as a basic reference standard when verifying the accuracy of the results extracted in this study. We focused on extreme precipitation-induced flood events and similarly constrained the results derived from the discharge data with extreme precipitation data. To ensure accuracy, we first selected the floods extracted from discharge stations with GFS greater than 0.5 for comparison. Due to discharge reflects the amount of water integrated over its entire contributing basin and contributing time (Yang et al., 2019), we combined the flood events obtained from each discharge station in time series to characterize the flood events in the 261 watershed (HydroSHEDS Basins Level 4 (Lehner and Grill, 2013; Lehner et al., 2008)). The accuracy index used for comparison in this study is the probability of detection (POD) (Yang et al., 2021), i.e., whether the floods extracted from the discharge data correspond to the flood days extracted in this study. Although flood extracted from discharge cannot guarantee that the surrounding land will experience flooding, the discharge data time series provide us with necessary data to support the reliability of the time series verification process.*

*Figure 3 showed the global distribution of discharge location and the goodness of flood separation. The data distributions in North America, South America, Europe and south-eastern Australia were relatively dense, while data were seriously missing in central, northern and eastern Asia. The areas with higher GFS were located in the eastern part of the United States and central Europe, and the stations recorded in these areas were relatively complete. The discharge stations with GFS above 0.5 accounted for 73.49%. Figure 4 showed the flood events in the level-4 river basins. We find that most flood events reflected by the discharge data are mainly located in the eastern and western United States, central South America, eastern Europe and New Zealand.*

*The POD calculation results are shown in Figure 4. The darker the color is, the higher the corresponding flood detection accuracy is. We found that the overall accuracy performed well; the detection accuracies obtained for the central and eastern parts of the United States, western South America, southern Africa and around Australia are relatively high. Figure 5 showed the histogram of 261 watersheds of level-4 basins and the percentage of river basins with POD values greater than or equal to 0.5 reached 62 %. This finding showed that our extracted flood days also reflected relatively high accuracies in comparison with flood events at river basins.*

[Figure]

Figure 6 GFS of 3408 discharge sites around the world. The slant line means no data.

[Figure]

Figure 7 Flood events distribution in the Level-4 basins. The slant line means no data.

[Figure]

Figure 8 POD values in the Level-4 basins. The slant line means no data.

[Figure]

Figure 9 The histogram of 261 level-4 basins.

32. Line 349. The estimation of POD is not clear. This is an important part of the study related to the evaluation of the accuracy of obtained results. And this part is given in vary loose and general manner. Line 366. The time series of what?

**Response:** We thank reviewer's suggestions. We have made a detailed description in Lines 274-280, Page 8 to make it clearer.

*In order to better compare the relationship between flood events (observed from DFO, MODIS and discharge) and flood days (derived from GRACE), we referred to the probability of detection (POD) index proposed by Yang et al. (2021) and made it more appropriate for our study.*

$$\text{POD} = \text{flood}_{GRACE-base} \ /(\text{flood}_{observed} + \text{flood}_{miss}) \qquad (2)$$

*Where $flood_{GRACE-based}$ means flood events identified by GRACE, $flood_{observed}$ means DFO-recorded flood events, MODIS-derived flood events or discharge-derived flood events. If each flood events with a three- or five-day buffer could cover the GRACE-based flood days, we consider it a $flood_{GRACE-base}$ event.*

We also re-written the Line 366 sentence in Lines 446-447, Page 18.

*This finding showed that our extracted flood days also reflected relatively good accuracies in comparison with flood events derived from discharge time series at same river basins.*

**Response:** We thank reviewer's suggestions. The "accuracy rate" means POD. We have re-written it in the manuscript in Lines 462-463, Page 19.

*This study compared the different PODs under different quantile threshold scenario when comparing with DFO database.*

34. Lines 376-378. I would be surprised if the "precipitation-type" floods are not related to the "extreme precipitation".

**Response:** We thank reviewer's suggestions. We have re-written it in Lines 463-466, Pages 19.

*Figure 16 shows that the selection of different thresholds in the two extreme precipitation scenarios influenced the flood extraction accuracy of POD with contributions ranging from 72.4% to 81.4%. This shows that the selected thresholds can affect the detection rate of approximately 9% (approximately 214) of flood events.*

**Response:** We thank reviewer's suggestions. We have re-written it in Lines 487-488, Page 20:

*To further verify the reliability of our GRACE-based flood products, we compared them with the flood events extracted from global GRDC discharge data, and the probability of detection greater than or equal to 0.5 reached 62% at river basin scale.*

**Response:** We thank reviewer's suggestions. We have re-written it in Lines 490-491, Page 20.

*These results also showed that our GRACE-based flood days could identify and supplement flood events not recorded by DFO.*

37. Line 405. The use "for the first time" is not correct as both DFO dataset and

MODIS- derived flood events product have global coverage.

**Response:** We thank reviewer's suggestions. This statement is inappropriate and we have re-written it in Lines 484-485, Page 20.

*This study successfully extracted global flood days using GRACE TWS and extreme precipitation data between 60°S and 60°N from Apr. 1st, 2002 to Aug. 31st, 2016.*

38. Line 411. Please, explain what is the value of the flood event product of such low (1°x1°) spatial resolution?   Discuss also the cases of false detection of flood events, i.e.  the events not supported by datasets/products used for verification of the obtained results. Provide their statistics.   How do you separate false flood detection from non recorded flood cases?

**Response:** We thank reviewer's suggestions. The GRACE-based flood days own wide coverage (covering between 60°S and 60°N). It is continuous in time and space and the number of flood days in different areas or in different research time scales can be calculated according to research needs, which makes up for the lack of flood events due to weather condition for MODIS and records missing for DFO. It provides not only important data support for the spatiotemporal distributions and attributions of global flood events, but also a reference for large-scale quasi-real-time flood event monitoring with the development of GRACE-FO and the quality improvement of GRACE daily data.

We agree reviewer's suggestions that false detection of flood events is important. However, due to the difficulty in observation, a complete global record of floods is unavailable. This also highlights the importance of this study which tries to provide a new approach for detecting global flood events. The most available reference observation datasets include DFO and GRDC discharge datasets. The number of flood events recorded by the DFO data is also incomplete, the global distribution of GRDC discharge data is uneven and some measurements are still missing in time series. Moreover, we couldn't correctly separate specific flood events from GRACE-based flood days, which is also direction that needs further study in the future. We therefore cannot calculate the false alarm rate based on the available incomplete dataset. However, we calculated the probability of detection (POD) of DFO (including MODIS-based flood inundation) and GRDC discharge based on existing observational datasets, and selected larger flood events recorded by news media (not recorded by DFO) for further comparison. Although we cannot calculate the false alarm rate, we can calculate the corresponding detection rate (i.e. POD) for the existing recorded floods. We also acknowledged that we could not effectively separate false flood detection and non-recorded flood cases. We have discussed these shortcomings in the discussion section.

We have supplemented the value of flood events in Lines 484-497, Page 20 and added these discussions in Lines 498-512, Page 21.

[revised manuscript text omitted]